# Claustrum and dorsal endopiriform cortex complex cell-identity is determined by Nurr1 and regulates hallucinogenic-like states in mice

Ioannis Mantas [1,2] ✉, Ivana Flais[1,3,4], Yuvarani Masarapu [5], Tudor Ionescu[3], Solène Frapard [5], Felix Jung[2], Pierre Le Merre [2], Marcus Saarinen[1], Katarina Tiklova[6], Behzad Yaghmaeian Salmani [6], Linda Gillberg[6], Xiaoqun Zhang [1], Karima Chergui[7], Marie Carlén [2], Stefania Giacomello [5], Bastian Hengerer [3], Thomas Perlmann[6] & Per Svenningsson [1]

The Claustrum/dorsal endopiriform cortex complex (CLA) is an enigmatic brain region with extensive glutamatergic projections to multiple cortical areas. The transcription factor Nurr1 is highly expressed in the CLA, but its role in this region is not understood. By using conditional gene-targeted mice, we show that Nurr1 is a crucial regulator of CLA neuron identity. Although CLA neurons remain intact in the absence of Nurr1, the distinctive gene expression pattern in the CLA is abolished. CLA has been hypothesized to control hallucinations, but little is known of how the CLA responds to hallucinogens. After the deletion of Nurr1 in the CLA, both hallucinogen receptor expression and signaling are lost. Furthermore, functional ultrasound and Neuropixel electrophysiological recordings revealed that the hallucinogenic-receptor agonists' effects on functional connectivity between prefrontal and sensorimotor cortices are altered in Nurr1-ablated mice. Our findings suggest that Nurr1-targeted strategies provide additional avenues for functional studies of the CLA.

Claustrum and dorsal endopiriform nucleus compose a complex (CLA), which is a sheet-like grey matter structure with a long rostrocaudal axis and lies between the caudoputamen and agranular insula[1]. CLA neurons exhibit an extraordinary broad axonal projection network that wraps around the entire ipsilateral cortical mass[2]. This feature led Francis C Crick and Christof Koch to hypothesize that the CLA is the main brain region that is responsible for perceived consciousness[3]. Nevertheless, there is a growing body of literature about CLA's function, which supports its importance in multiple aspects, such as the control of attention, slow wave sleep, depressive-like behavior, and pain processing[4,5]. The highest density of CLA efferent arborizations is found in PFC structures[6,7]. CLA projection neurons are glutamatergic, and they target both inhibitory and excitatory neurons in the cortex[7]. Some studies report that in certain cortical areas, CLA neurons primarily project to parvalbumin and neuropeptide-Y GABAergic interneurons[8]. Due to its broad cortical projection pattern, the CLA has been hypothesized as the master region that orchestrates the hallucinogenic experience[9]. Supporting

[1]Department of Clinical Neuroscience, Karolinska Institutet, Stockholm, Sweden. [2]Department of Neuroscience, Karolinska Institutet, Stockholm, Sweden. [3]CNSDR, Boehringer Ingelheim Pharma GmbH & Co. KG, Biberach, Germany. [4]Department of Neuroimaging King's College London, London, UK. [5]Science for Life Laboratory, Department of Gene Technology, KTH Royal Institute of Technology, Stockholm, Sweden. [6]Department of Cell and Molecular Biology, Karolinska Institutet, Stockholm, Sweden. [7]Department of Physiology and Pharmacology, Karolinska Institutet, Stockholm, Sweden. ✉e-mail: ioannis.mantas@ki.se

this hypothesis, there is evidence that CLA neurons display robust expression of receptors for psychedelics and opioidergic hallucinogens, such as the 5-hydroxytryptamine receptor 2 A (5HT$_{2A}$R), 5-hydroxytryptamine receptor 2 C (5HT$_{2C}$R), and kappa opioid receptor (κOR)[10–13]. All these receptors mediate the effects of psychedelics and opioidergic hallucinogens[10,14]. Therefore, the CLA's connectivity and abundance of hallucinogen receptors make it a suitable structure to mediate the brain's effects of hallucinogens.

Apart from the high levels of 5HTRs and κOR, CLA is characterized by the high expression levels of Nur receptor-related 1 (Nurr1), which is also named *Nr4a2*[15]. Even though Nurr1 is a well-established marker for the identification of CLA cells, its role in this region is still obscure[15,16]. Nurr1 is a transcription factor known to be required for midbrain dopamine (DA) neuron development[17] and regulation genes that characterize the dopaminergic molecular phenotype[17]. Nurr1 is believed to function as a monomer but can also exert effects through heterodimer formation together with the retinoid X receptor[18]. Mutations in the Nurr1 gene in humans have been linked to Dopa-Responsive Dystonia Parkinsonism[19].

To explore the role of Nurr1 in the CLA, we generated conditional knock-out (KO) mice lacking Nurr1 in CLA neurons. Given the hypothesis that the CLA is crucial for distinct vigilance states, we utilized these mice to investigate the imaging and electrophysiological correlates of human hallucinogenic-like states. Specifically, we observed the reduced cerebral blood flow in the PFC, heightened functional connectivity (fc) indicated by blood-oxygen-level-dependent (BOLD) signals among neural networks, and diminished global coherence in electroencephalography (EEG)[20–25]. These parameters have been associated with hallucinatory experiences in humans, are here assessed in the Nurr1-deficient CLA mice.

## Results

### CLA Nurr1 expression is affected by Nurr1 gene dosage

Even though Nurr1's role in DA neurons has been extensively studied, the functional importance of the transcription factor in CLA is still unknown[26]. First, we used in situ hybridization (ISH) to detect Nurr1 mRNA in CLA of various mammalian species including human (Fig. 1a, b). Strong density of Nurr1 transcripts was confined between the insular and striatal regions and was conserved in both primates (*Homo sapiens, Callithrix jacchus*) and rodents (*Rattus norvegicus, Mus musculus*) (Fig. 1b). Then we aimed to investigate whether Nurr1 gene expression in CLA is differentially regulated compared to other Nurr1 enriched brain regions[27,28]. Therefore, we performed ISH experiments in Nurr1-heterozygous-KO mice (Fig. 1c). Interestingly, these experiments showed that Nurr1 transcript density in CLA, layer 6b (L6b) and subiculum are proportional to the Nurr1 gene copy number, while the hippocampal and ventral midbrain Nurr1 mRNA levels appear to compensate for the loss of one Nurr1-allele (Fig. 1d). Thus, we provide evidence that Nurr1 might play a different role in the CLA compared to other known Nurr1-rich brain regions, such as DA neurons and the hippocampus.

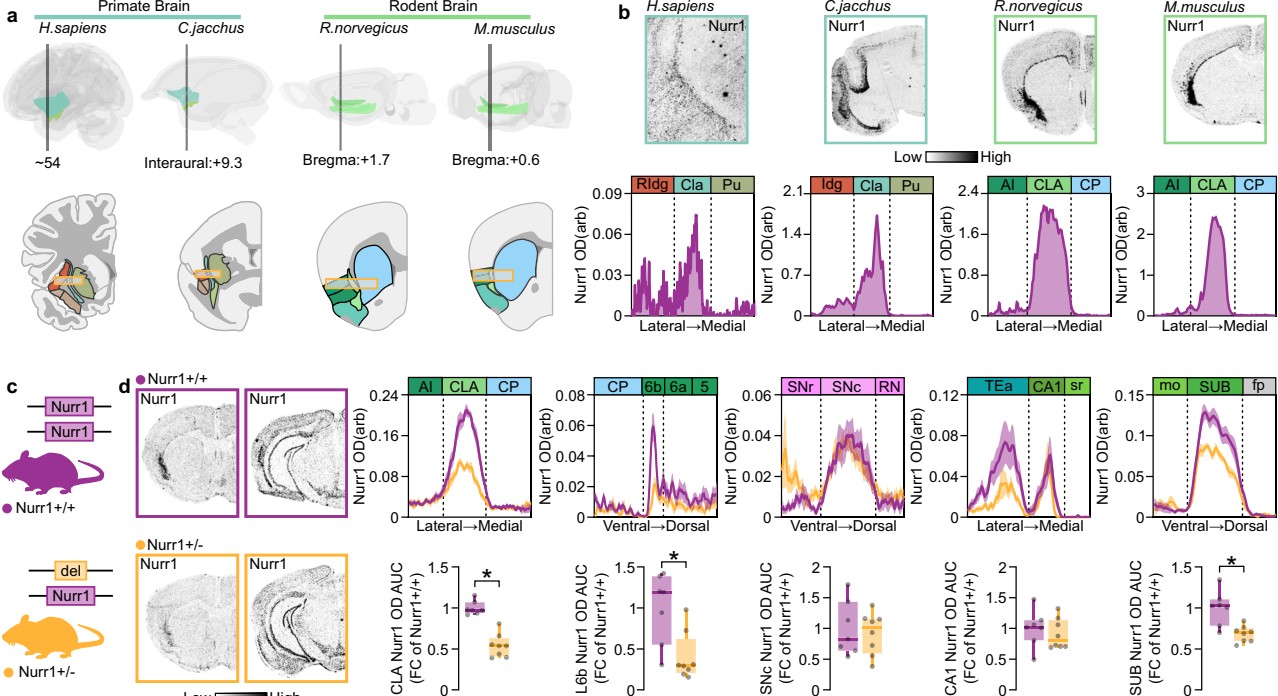

**Fig. 1 | CLA Nurr1 mRNA is conserved among species and is dependent on Nurr1 gene dosage. a** Schematic depiction of the CLA in primate (H. sapiens and C. jacchus) and rodent (R. norvegicus and M. musculus) brain and its location between the Insular cortex and Striatum. Yellow framed rectangles are centered on CLA and delineate the axis across the average OD traces were measured **b** Nurr1 mRNA is enriched in human (H. sapiens), marmoset (C. jacchus), rat (R. norvegicus) and murine (M. musculus) CLA. **c** Illustration of Nurr1+/− mouse line. **d** Left: ISH autoradiographs showing Nurr1 mRNA at CLA and SNc level. Right: Line-graphs (upper) and boxplots (lower) showing the Nurr1 mRNA OD quantification in CLA, L6b, SNc, CA1, and SUB of Nurr1+/+ and Nurr1+/− mice (Nurr1+/+: n = 7, Nurr1+/−: n = 8; CLA: *p < 0.0001, L6b: *p = 0.0067, SUB: *p = 0.0021, Two-sided Unpaired *t*-test). Data in line-graphs are expressed as mean ± SEM. Boxplots show all data

points, the 25th and 75th percentile (box), the median (center), and the maxima (whiskers). OD: optic density, arb: arbitrary units, RIdg: rostral dysgranular insular cortex, Idg: insular dysgranular cortex, Cla: human claustrum/dorsal endopiriform cortex complex/dorsal endopiriform cortex complex, Pu: putamen, AI: agranular insula, CLA: rodent claustrum/dorsal endopiriform cortex complex/dorsal endopiriform cortex complex, CP: caudoputamen, AUC: area under the curve, FC: fold change, L6b: cortical layer 6b, SNr: substantia nigra reticular part, SNc: substantia nigra compact part, RN: reticular nucleus, TEa: temporal association areas, sr: stratum radiatum of CA1, mo: molecular layer of dentate gyrus, SUB: subiculum, fp: posterior forceps of corpus callosum. ISH: in situ hybridization. Source data are provided as a Source Data file.

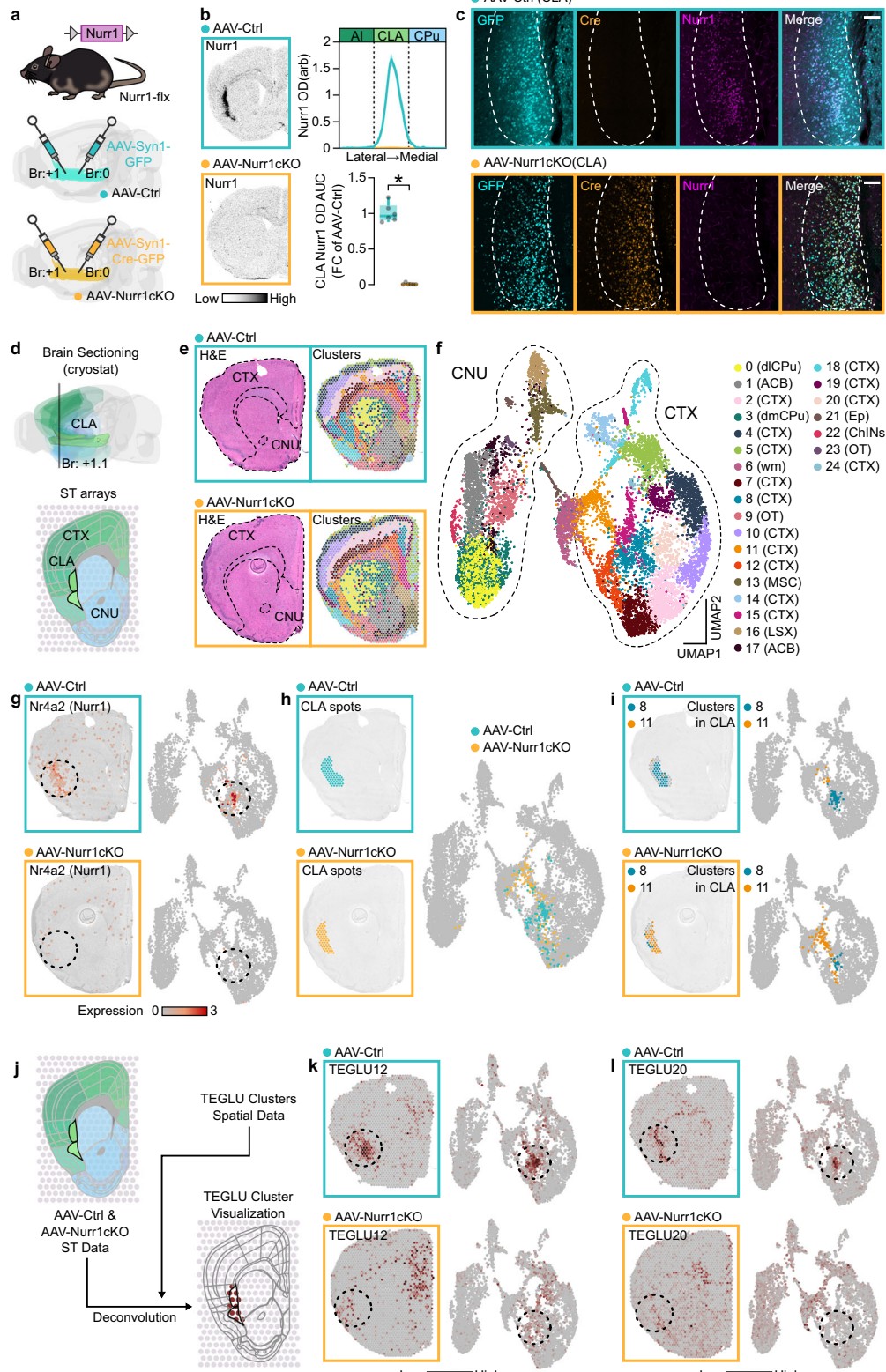

## Nurr1 loss alters the transcriptomic profile of the CLA

Deletion of Nurr1 from both alleles leads to lethality in early postnatal life[17]. Thus, to investigate Nurr1's transcriptional role in CLA we injected adeno-associated viruses (AAVs) that carried transgenes encoding either green fluorescent protein (GFP) (AAV-Ctrl) or Cre fused with GFP (AAV-Cre) in the CLA of conditional Nurr1 gene-targeted mice (Nurr1-flx) (Fig. 2a). The resulting mice lacked Nurr1 mRNA and protein in CLA

of AAV-Nurr1cKO mice as shown by radioactive ISH and immunohistochemsitry (Fig. 2b, c). To analyze the consequences of Nurr1 ablation on genome-wide CLA gene expression, sections at CLA level of these mice were used in spatial transcriptomic (ST) arrays (Fig. 2d, e). Analysis of ST gene expression data using dimensionality reduction and clustering separated the spots in 25 different clusters (Fig. 2e, f, Supplementary Fig.1, Supplementary Data 1, Supplementary Data 2). The

**Fig. 2 | Nurr1 loss changes the transcriptomic profile of the CLA. a** Illustration of Nurr1-flx mouse line construct and the viral strategy of Nurr1 deletion in CLA. **b** ISH autoradiographs (left) and graphs (right) showing Nurr1 mRNA at the CLAs of AAV-Ctrl and AAV-Nurr1cKO mice. (AAV-Ctrl: $n = 6$, AAV-Nurr1cKO: $n = 6$; *$p < 0.0001$, Two-sided Unpaired $t$-test. **c** Immunofluorescence showing GFP, Cre, and Nurr1 expression in the CLAs of AAV-Ctrl (upper) and AAV-Nurr1cKO (lower) mice (scale-bar: 200 μm). **d** Illustration showing the level and the brain structures of the coronal sections that were hybridized on ST arrays. **e** H&E-stained brain sections (left) and on-tissue visualization of the 25 molecular clusters in AAV-Ctrl and AAV-Nurr1cKO brains (AAV-Ctrl: $n = 3$, AAV-Nurr1cKO: $n = 3$). **f** 2D-UMAP with a categorical color code to visualize the spots' distribution in the 25 identified molecular clusters. **g** On-tissue (left) and 2D-UMAP (right) visualization of Nurr1 expression in AAV-Ctrl and AAV-Nurr1cKO sections. **h** On-tissue (left) and 2D-UMAP (right) visualization of the spots located in the CLA region from AAV-Ctrl and AAV-Nurr1cKO sections. **i** On-tissue (left) and 2D-UMAP (right) visualization of clusters 8 and 11 in CLA region from AAV-Ctrl and AAV-Nurr1cKO sections. **j** Schematic depiction of the deconvolution analysis using the dataset from Zeisel et al. 2018. **k, l** On-tissue (left) and 2D-UMAP (right) visualization of TEGLU12 (**k**) and TEGLU20 (**l**) from AAV-Ctrl and AAV-Nurr1cKO sections. Data in line-graphs are expressed as mean ± SEM. Boxplots show all data points, the 25th and 75th percentile (box), the median (center), and the maxima (whiskers). The dashed circle denotes the CLA region. AAV: adeno-associated virus, ISH: in situ hybridization, GFP: green fluorescent protein, OD: optic density, arb: arbitrary units, AI: agranular insula, CLA: claustrum/dorsal endopiriform cortex complex, CP: caudoputamen, AUC: area under the curve, FC: fold change, ST: spatial transcriptomics, CTX: cerebral cortex, CNU: cerebral nuclei, H&E: hematoxylin & eosin staining, dlCP: dorsolateral caudoputamen, dmCP: dorsomedial caudoputamen, ACB: nucleus accumbens, wm: white matter, OT: olfactory tubercle, MSC: medial septal complex, LSX: lateral septal complex, Ep: ependymal cells, ChINs: striatal cholinergic interneurons, TEGLU: telencephalon excitatory projecting neurons. Source data are provided as a Source Data file.

major diversity seen in UMAP space is the division between spots in the cerebral nuclei or cortical plate region (Fig. 2f). Both the on-tissue location and the significantly enriched genes of each cluster allowed us to identify clusters that define the different cortical layers and subplate, dorsolateral and dorsomedial CP, nucleus accumbens, striatal cholinergic interneurons, olfactory tubercle, septal region, ependymal cells and white matter region (Fig. 2f, Supplementary Data 2). Nurr1 expression visualized in the UMAP diagram indicated high expression in spots representing the CLA region of AAV-Ctrl mice but, as expected, was absent in AAV-Nurr1cKO mice (Fig. 2g). To characterize the changes that take place after the loss of Nurr1, we attempted to visualize the spots located in CLA region on UMAP (Fig. 2g). Interestingly, CLA spots from AAV-Ctrl and AAV-Nurr1cKO mice occupied different space in the ST UMAP (Fig. 2h). On-tissue visualization of the clusters demonstrated that cluster 8 spots localized to the CLA region in AAV-Ctrl mice while cluster 11 spots corresponded to the CLA region in AAV-Nurr1cKO mice (Fig.2i). These clusters are clearly distinct in the UMAP visualization which indicates a robust change in gene expression as a consequence of Nurr1 ablation in this region. For further confirmation of the CLA transcriptomic profile loss, we proceeded to deconvolution of our ST data by using the telencephalic-excitatory-neuron-specific (TEGLU) gene expression information from Zeisel et al. [29] (Fig. 2j). In this dataset, TEGLU12 and TEGLU20 clusters appear specific for CLA region. Visualization of TEGLU12 and TEGLU20 in AAV-Ctrl sections showed the CLA-specific localization of these clusters (Fig. 2k, l). However, this CLA-specific representation of these clusters was substantially lost in AAV-Nurr1cKO brains (Fig. 2k, l). On-tissue and UMAP visualization of other neighboring TEGLU clusters, such as TEGLU2 (L6b), TEGLU3 (L6a) and TEGLU10 (L5), did not show any obvious changes between AAV-Ctrl and AAV-Nuur1cKO mice (Supplementary Fig. 2).

## Nurr1 deletion causes downregulation of CLA-enriched genes

To further characterize the transcriptional changes that occur after Nurr1 deletion in CLA, differentially expressed (DE) genes in the two conditions were identified (Fig. 3a, Supplementary Data 3). Of note, several genes that are specifically enriched in CLA (according to inspection of genes in Allen Brain Anatomic Gene Expression Atlas and Erwin et al. [13]) were significantly downregulated in AAV-Nurr1cKO mice, including *Gnb4*, *Gng2*, *Rgs12* and *Oprk1* (κOR) (Supplementary Fig. 3). Additionally, several CLA-specific genes were enriched in cluster 8 but diminished in cluster 11 (Fig. 3b, Supplementary Fig. 3, Supplementary Data 2). Both on-tissue and UMAP visualization of the spots showed notable expression reduction of several CLA-enriched genes (Fig. 3c–f, Supplementary Fig. 4). Radioactive ISH showed robust downregulation in AAV-Nurr1cKO mice of several of these CLA markers (*Gnb4*, *Sstr2*, *Ntng2*, *Slc17a6*, *Bdnf* and *Oprk1* (κOR)) (Fig. 3g–l, Supplementary Fig. 3). κOR is a well-known hallucinogen receptor, and it has been reported

that other hallucinogen receptors including 5HT$_{2A}$R and 5HT$_{2C}$R are also enriched in the CLA. All three genes are downregulated in AAV-Nurr1cKO mice as revealed by ISH (Fig. 3m, n). However, all these experiments focused on the detection of mRNA changes induced by Nurr1 deletion. Therefore, we wanted to address if these effects extend to a protein level. In line with the mRNA results, immunofluorescent experiments in AAV-Nurr1cKO brains showed notable reduction of 5HT$_{2A}$R signal in CLA (Supplementary Fig. 5).

## Claustrocortical projections are retained after conditional deletion of Nurr1 in CLA

Due to the profound changes in many CLA-enriched genes, we questioned if Nurr1 deletion affects the integrity of CLA-cortical projection neurons. To assess this question, we injected retrograde AAVs (rgAAVs) that carried transgenes encoding Cre (rgAAV-Cre) in the PFC of both wild-type (WT) and Nurr1-flx mice (Supplementary Fig. 6). At the same time, we injected an AAV that carried a Cre-dependent GFP transgene (DIO-GFP) in CLA of the ipsilateral hemisphere (Supplementary Fig. 6). RgAAV-Cre produced a reduction of Nurr1 CLA expression in Nurr1-flx mice (Supplementary Fig. 6). However, both WT and Nurr1-flx mice displayed GFP labeled cells in CLA, providing evidence that Nurr1 ablation might not interrupt CLA-cortical axon integrity (Supplementary Fig. 6). Recently, CLA-cortical arborizations were reported to develop during early postnatal period[30]. Moreover, prenatal deletion of Nurr1 leads to midbrain DA neuron agenesis and disruption of the nigrostriatal pathway[17,31]. Thus, we investigated the possibility that Nurr1 deletion causes agenesis of CLA neurons. Since DA receptor D1 (D1R) is reported to be expressed in CLA, we aimed to use D1R-Cre mice to knock out Nurr1 early in CLA neurons while maintaining Nurr1 expression intact in most brain regions that do not express D1R (SNc/VTA). By using both FISH (RNAscope) and genetic labeling of D1R-positive cells with a D1R-Cre/mCherry-flx mouse line, we observed that 85% of CLA Nurr1+ cells co-express D1R mRNA (Fig.4a–d). Immunofluorescence experiments in postnatal day 1 (PND1) brains from conditional D1R-Cre/Nurr1-flx mouse line (D1R-Nurr1cKO) revealed a decrease in CLA Nurr1 expression (Supplementary Fig. 7). ISH in adult brains from the same mouse line showed a robust (96%) Nurr1 mRNA reduction in the CLA (Fig. 4e, f). In contrast, Nurr1 expression in CA1 was maintained (Fig. 4f). L6b, SNc, and SUB Nurr1 expression were significantly reduced, but to a lesser extent than CLA. (Fig. 4f). Radioactive ISH was used to detect a deficient non-coding transcript that is maintained after Cre-mediated Nurr1 ablation (Supplementary Fig. 7). This transcript was detected in the CLA of both Ctrl and in D1R-Nurr1cKO mice indicating that depletion of Nurr1 protein does not lead to CLA neuron loss (Supplementary Fig. 7). Further experimentation with unilateral injections of fluorogold (Fg) into the PFC (anterior cingulate area-secondary motor area) resulted in retrograde labeling of the ipsilateral CLA in both Ctrl and

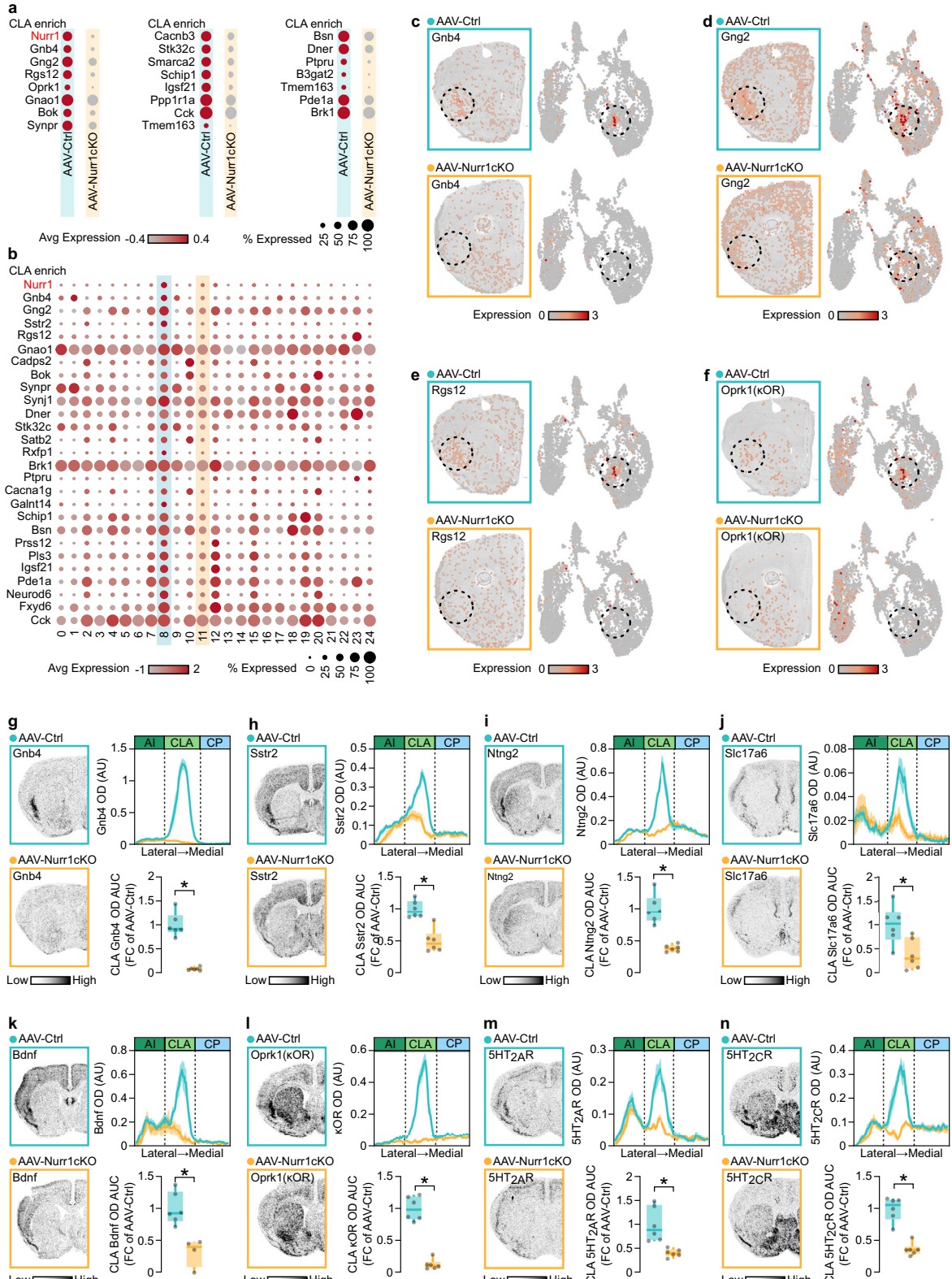

D1R-Nurr1cKO (Fig. 4g, h). Additional injections of retrograde AAVs that contain Tdtomato (rgAAV-Tdt), into PFC, led to retrograde labeling of the ipsilateral CLA, regardless of genotype (Supplementary Fig. 7). Together these neuronal tracing strategies indicate that CLA-PFC projections are unaffected by Nurr1 ablation.

CLA dysfunction has been associated with cognitive control, attentional functions, depression-like behavior, and pain-predictive cues[32–34]. Thus, we assessed the performance of D1R-Nurr1cKO mice in

a broad suite of behavioral tests that examine these behavioral aspects. D1R-Nurr1cKO mice' performance in the Y-maze spontaneous alternation test and passive avoidance task indicate that Nurr1 loss in CLA does not produce perseveration or pain-avoidance learning deficits (Fig. 4i, j). To assess auditory attention of D1R-Nurr1cKO mice, we used a modified behavioral paradigm from Atlan et al.[35], in which we examined mice' feeding motivation in the presence of an auditory distractor (Fig. 4k). Interestingly, we did not observe any significant

**Fig. 3 | Loss of Nurr1 in CLA reduces the expression of CLA-enriched genes. a** Dot plot showing the expression of the differentially expressed CLA-enriched genes between AAV-Ctrl and AAV-Nurr1cKO CLA spots. **b** Dot plot showing the expression of the CLA-enriched genes that were differentially expressed in cluster 8. **c–f** On-tissue (left) and 2D-UMAP (right) visualization of Gnb4 (**c**), Gng2 (**d**), Rgs12 (**e**), Oprk1 (κOR) (**f**) expression in AAV-Ctrl and AAV-Nurr1cKO sections. **g–n** ISH autoradiographs (left) and graphs (right) showing Gnb4 (**g**), Sstr2 (**h**), Ntng2 (**i**), Slc17a6 (**j**), Bdnf (**k**), Oprk1 (κOR) (**l**), 5HT$_{2A}$R (**m**) and 5HT$_{2C}$R (**n**) mRNA at CLA of AAV-Ctrl and AAV-Nurr1cKO mice (AAV-Ctrl: $n = 6$, AAV-Nurr1cKO: $n = 6$, AAV-

Nurr1cKO in **k**: $n = 4$; **g**: $*p < 0.0001$, **h**: $*p = 0.0003$, **i**: $*p < 0.0001$, **j**: $*p = 0.0147$, **k**: $*p = 0.0022$, **l**: $*p < 0.0001$, **m**: $*p = 0.0027$, **n**: $*p < 0.0001$, Two-sided Unpaired *t*-test). Data in line-graphs are expressed as mean ± SEM. Boxplots show all data points, the 25th and 75th percentile (box), the median (center), and the maxima (whiskers). The dashed circle denotes CLA region. Avg: average, CLA: claustrum/dorsal endopiriform cortex complex, OD: optic density, arb: arbitrary units, AI: agranular insula, CP: caudoputamen, AUC: area under the curve, FC: fold change, ISH: in situ hybridization. Source data are provided as a Source Data file.

difference between genotypes in both milk consumption and feeding latency at the presence of the distructor (Fig. 4k). Moreover, we showed that D1R-Nurr1cKO mice do not display any pre-pulse inhibition deficit, which underscores the intact auditory attention of this mouse line (Fig. 4l). Further experimentation with rodent paradigms of depression-like behavior showed that D1R-Nurr1cKO mice exhibit unaltered performance in sucrose preference test (SPT) and forced swim test (FST) (Fig. 4m, n).

**D1R-Nurr1cKO mice lose their CLA transcriptional identity**

To extend the findings obtained from AAV-Nurr1cKO mice, we next investigated if D1R-Nurr1cKO mice exhibit an altered CLA transcriptomic profile. Bulk RNA-sequencing was performed after the dissection of the area between the striatum and insula (Fig. 5a). DE analysis showed that D1R-Nurr1cKO mice exhibit a significant downregulation of several genes that were found enriched in cluster 8 from the ST experiment described above (Fig. 2). These genes include *Cadps2*, *Synj1*, *Rgs12*, *Gnao1*, *Neurod6*, *Gng2* and *Sstr2* (Fig. 5b, c). Moreover, the same analysis revealed that D1R-Nurr1cKO mice display significant downregulation of *Ntng2* and *Sema5b*, which are CLA-enriched genes according to AGEA and Erwin et al.[13](Fig. 5b, c, Supplementary Fig. 3). Like AAV-Nurr1cKO mice, radioactive ISH experiments revealed significant reduction of the CLA markers *Gnb4*, *Sstr2*, *Ntng2*, *Slc17a6* and *Bdnf* in D1R-Nurr1cKO mice (Fig. 5e-n). In addition, similar to the AAV-Nurr1cKO adult mice, CLA-enriched hallucinogen receptors, 5HT$_{2A}$R, 5HT$_{2C}$R, and κOR were also reduced in the D1R-Nurr1cKO mice (Fig. 5o-t). Since D1R-Nurr1cKO mice display a notable reduction of Nurr1 in L6b (Fig. 4d), we aimed to examine the integrity of L6b markers in these mice. Consistent with previous studies, our radioactive ISH demonstrated that apart from CLA, 5HT$_{2A}$R is enriched in L6b as well (Supplementary Fig. 8). Nevertheless, the same experiments revealed that there is no significant change of L6b 5HT$_{2A}$R mRNA density between Ctrl and D1R-Nurr1cKO brains (Supplementary Fig. 8). *Tmem163* is considered a common marker of CLA and L6b. Radioactive ISH with *Tmem163* anti-sense riboprobe showed that D1R-Nurr1cKO mice display a strong reduction of *Tmem163* mRNA in CLA but not in L6b, indicating the preservation of L6b molecular identity in this mouse line (Supplementary Fig. 8). These results, together with the postnatal virus-mediated deletion of Nurr1 in the CLA, indicate a pivotal role for Nurr1 in maintaining the transcriptional identity of the CLA neurons.

**Deletion of Nurr1 in CLA abolishes hallucinogen receptors' effects in CLA**

It is well known that the 5HT$_{2A}$R and 5HT$_{2C}$R are primarily coupled to Gα$_q$ subunits, while the κOR mainly recruits Gα$_i$. As a result, we examined both 5HT$_{2A}$R and κOR agonist effects in the CLAs of D1R-Nurr1cKO mice. 2,5-dimethoxy-4-iodoamphetamine (DOI) is a specific 5HT$_2$R agonist that has been described to increase immediate early gene transcripts in several brain regions such as the layer 5 of primary somatosensory cortex (SSp) and CLA[36,37]. Systemic DOI (8 mg/kg) administration upregulated both *Fos* and *Arc* in layer 5 of SSp in both Ctrl and D1R-Nurr1cKO (Supplementary Fig. 9). We also observed that DOI increased the expression of *Fos* and *Arc* in CLAs of Ctrl mice (Fig.

6a–c, Supplementary Fig. 9). However, the effect was abolished in D1R-Nurr1cKO mice (Fig. 6b, c). Apart from *Fos* upregulation, DOI has been reported to cause internalization of 5HT$_{2A}$R in neurons[37]. Therefore, we desired to examine this effect in D1R-Nurr1cKO mice treated with DOI. Similar to AAV-Nurr1cKO mice, saline-treated D1R-Nurr1cKO mice display a notable reduction of the 5HT$_{2A}$R immunoreactive neuropil in CLA (Fig. 6d). DOI produced a clear relocation of 5HT$_{2A}$R in the cytoplasm of CLA neurons in Ctrl mice (Fig. 6e). This effect was markedly reduced in D1R-Nurr1cKO mice (Fig. 6e). 5HT$_{2A}$R-based psychedelics administration causes a "head twitch response" (HTR) in rodents, which is dependent on the 5HT$_{2A}$R expression in the forebrain[36]. D1R-Nurr1cKO mice did not display diminished head twitches after the administration of 8 mg/kg or 1 mg/kg of DOI but rather increased HTR counts compared to the Ctrl mice at the dose of 8 mg/kg (Supplementary Fig. 9). To investigate if this effect arises from loss of 5HT$_{2A}$R or 5HT$_{2C}$R in CLA, we examined the HTR in AAV-Nurr1cKO mice as well (Supplementary Fig. 9). These mice did not exhibit any significant difference in HTR compared to AAV-Ctrl, indicating that the increased HTR in D1R-Nurr1cKO mice results from Nurr1 ablation outside of the CLA (Supplementary Fig. 9). Considering the strong κOR expression in the CLA, we hypothesized that the receptor's Gα$_i$ activity[14] would hamper the region's response to excitatory afferents (Fig. 6f). Thus, we used U-69,593 (U69), a κOR agonist that displays similar signaling properties as the hallucinogen salvinorin, but with higher selectivity for κOR[38,39]. Accordingly, slice electrophysiology recordings showed that U69 reduced field excitatory postsynaptic potentials (fEPSP) magnitude in CLAs of Ctrl mice (Fig. 6f–h). However, this effect was abolished in the CLAs of D1R-Nurr1cKO mice (Fig. 6g, h). In conclusion, deletion of Nurr1 leads to loss of 5HT$_2$R and κOR expression in the CLA, which leads to abolished gene transcription and electrophysiological responses to their ligands in this brain area.

Additional to hallucinations, CLA has been suggested that plays a role in the transcriptional and behavioral effects of dopaminergic psychostimulants, like cocaine[40]. Therefore, we aimed to investigate the psychostimulants' induced *Arc* upregulation and hyperlocomotion. First, we examined the expression of D1R in the CLAs of D1R-Nurr1cKO mice (Supplementary Fig.10). In contrast to all the other genes that were examined in this study, D1R mRNA remained unchanged in the CLAs of D1R-Nurr1cKO mice (Supplementary Fig. 10). However, 7 days treatment of 10 mg/kg cocaine did not cause any significant upregulation of *Arc* in CLA (Supplementary Fig. 10). The same group of mice displayed a significant treatment effect on *Arc* expression in dorsomedial CP independent of the genotype (Supplementary Fig. 10). Before assessing the D1R-Nurr1cKO mice' hyperlocomotion response to psychostimulants, we first examined their baseline open field test (OFT) activity (Supplementary Fig. 10). D1R-Nurr1cKO mice demonstrated an increased average speed in OFT (Supplementary Fig. 10). Because this effect was not present in AAV-Nurr1cKO mice, we concluded that the observed OFT hyperactivity of D1R-Nurr1cKO mice might be a CLA independent off-target effect (Supplementary Fig. 10). It has been shown that the inhibition of CLA through virally induced expression of potassium channels reduces the locomotor response to 20 mg/kg cocaine[40]. Even though 7 days of treatment with medium cocaine (10 mg/kg) produced a robust

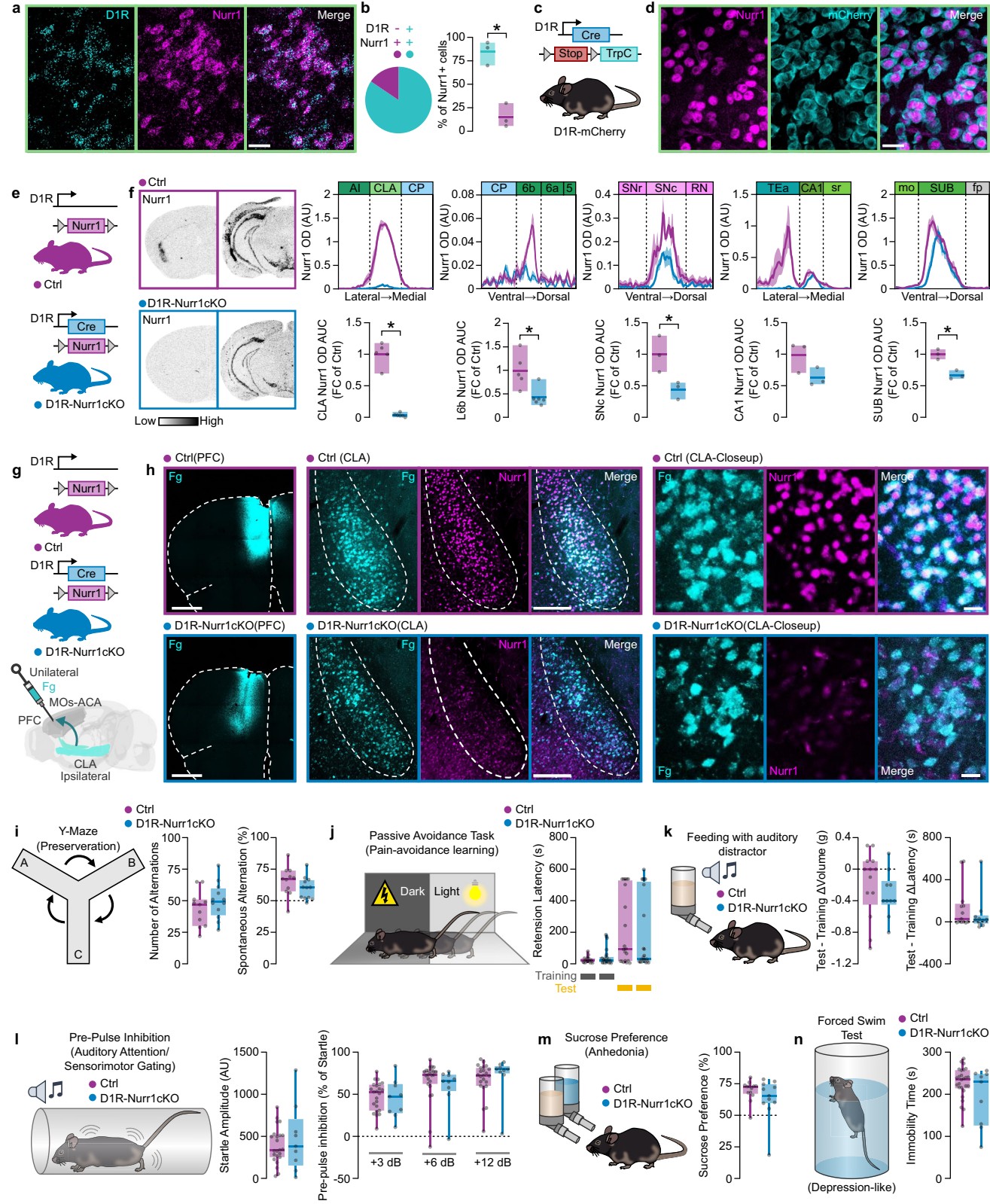

increase in locomotion, we did not observe any significant difference between Ctrl and D1R-Nurr1cKO mice (Supplementary Fig. 10). We also tested the acute locomotor response of D1R-Nurr1cKO mice to other psychostimulants, such as amphetamine (AMPH, 5 mg/kg) and phencyclidine (PCP, 8 mg/kg) (Supplementary Fig. 10). We did not observe any substantial genotype difference after the administration of AMPH or PCP (Supplementary Fig. 10).

## Nurr1 deletion in the CLA alters cortical functional connectivity in response to hallucinogen receptor agonists

Reduced cerebral blood flow in the PFC, increased BOLD-fc between default mode network (DMN) and sensorimotor (SM) network, and global desynchronization in (EEG) have been observed to correlate with the occurrence of hallucinatory experiences in humans[21,22]. For this reason, we used pharmacological functional ultrasound (pfUS), a

**Fig. 4 | Loss of Nurr1 in CLA does not affect claustrocortical projections.**
**a** RNAscope images showing cells that co-express Nurr1 and D1R mRNA in murine CLA (scale-bar: 30 μm). **b** Pie-chart and bar-graph showing the percentage of Nurr1/D1R double-positive cells (*p = 0.0027, Two-sided Paired t-test). **c** Illustration of the D1R-mCherry mouse-line construct. **d** Immunofluorescent images showing Nurr1/TrpC double-positive cells in CLA (scale-bar: 30 μm). **e** Illustration of D1R-Nurr1cKO mouse strain construct. **f** Left: ISH autoradiographs showing Nurr1 mRNA at CLA and SNc level. Right: Line-graphs (upper) and bar graphs (lower) showing the Nurr1 mRNA OD quantification in CLA, L6b, SNc, CA1, and SUB (Ctrl-CLA and Ctrl-L6b: n = 5, D1R-Nurr1cKO-CLA and D1R-Nurr1cKO-L6b: n = 6, Ctrl-SNc, Ctrl-CA1, and Ctrl-SUB: n = 3, D1R-Nurr1cKO-SNc, D1R-Nurr1cKO-CA1, and D1R-Nurr1cKO-SUB: n = 3; CLA: *p < 0.0001, L6b: *p = 0.0002, SNc: *p = 0.0384, SUB: *p = 0.0055, Two-sided Unpaired t-test). **g, h** Illustration (**g**) and fluorescent images (**h**) showing unilateral retrograde tracing of CLA with Fg (left scale bar: 1 mm, central scale-bar: 200 μm, right scale-bar: 20 μm). **i–n** Illustration of behavioral task (left) and boxplots (right) showing mice' performance at Y-maze (**i**) (Ctrl: n = 11, D1R-Nurr1cKO: n = 12), passive avoidance task (**j**) (Ctrl: n = 20, D1R-Nurr1cKO: n = 25), auditory destruction task auditory destruction task (**k**) (Ctrl: n = 13, D1R-Nurr1cKO: n = 10), pre-pulse inhibition task (**l**) (Ctrl: n = 23, D1R-Nurr1cKO: n = 9), sucrose preference task (**m**) (Ctrl: n = 13, D1R-Nurr1cKO: n = 11) and forced swim test (**n**) (Ctrl: n = 26, D1R-Nurr1cKO: n = 9). Data in line-graphs and bar graphs are expressed as mean ± SEM and mean ± minima/maxima respectively. Boxplots show all data points, the 25th and 75th percentile (box), the median (center), and the minima/maxima (whiskers). OD: optic density, arb: arbitrary units, AI: agranular insula, CLA: rodent claustrum/dorsal endopiriform cortex complex, CP: caudoputamen, AUC: area under the curve, SNr: substantia nigra reticular part, RN: reticular nucleus, TEa: temporal association areas, sr: stratum radiatum of CA1, mo: molecular layer of dentate gyrus, SUB: subiculum, fp: posterior forceps of corpus callosum. ISH: in situ hybridization, MOs: secondary motor area, ACA: anterior cingulate area, Fg: fluorogold, rgAAV: retrograde adeno-associated virus, Tdt: tdtomato. Source data are provided as a Source Data file.

technology that allows greater sensitivity than fMRI in such alterations[41], to measure these parameters in the Nurr1-ablated mice. Because 5HT$_{2A/C}$R and κOR agonists affect vigilance states, we modified our sedation protocol by adding 0.4% isoflurane to dexmedetomidine (Supplementary Fig.11).

According to human studies, we aimed to detect cerebral blood volume (CBV) and fc changes in PFC (anterior cingulate area and secondary motor area; rodent equivalent of DMN[42]) and sensorimotor (SM) (primary motor area and primary somatosensory area; rodent equivalent of SM[42]) networks after the administration of selective 5HT$_2$R and κOR agonists (Fig. 7a, b)[22,25,43]. Systemic administration of DOI (1 mg/kg) decreased the PFC CBV signal, particularly in the MOs, in both Ctrl and D1R-Nurr1cKO mice. (Fig. 7c–e). This effect was diminished in mice pretreated with the 5HT$_2$R antagonist, ketanserin (Supplementary Fig. 11). Dynamic pfUS-fc analysis revealed a DOI-induced increase in correlation of the examined cortical areas with the rest of the cortex (Fig. 7f). Dynamic pfUS-fc analysis demonstrated an increase in CBV-signal correlation between PFC and SM of Ctrl mice compared to baseline (Fig. 7g). These effects were absent in both D1R-Nurr1cKO mice and mice pretreated with ketanserin (Fig. 7f, g, Supplementary Fig. 11). Moreover, the area under the curve of dynamic pfUS-fc was significantly different between Ctrl and D1R-Nurr1cKO (Fig. 7g). Similar to DOI, U69 administration evoked a significant reduction in PFC CBV signal, especially in MOs, which was not different between the two genotypes (Fig. 7h–j). This effect was diminished in animals pretreated with the κOR antagonist, nor-binaltorphimine (nBNI) (Supplementary Fig. 11). Dynamic pfUS-fc analysis demonstrated a significant U69-induced increase in r-coefficient values of Ctrl mice compared to baseline, which was not present in D1R-Nurr1cKO mice (Fig. 7k, l). This U69-induced effect was absent in mice pretreated with nBNI as well (Supplementary Fig. 11). Furthermore, the area under the curve (AUC) of dynamic pfUS-fc between PFC and SM was significantly different between the two genotypes (Fig. 7l). These results indicate that 5HT$_2$R and κOR agonists share the ability to reduce PFC CBV and increase pfUS-fc between PFC and SM regions. With evidence suggesting a reduction in global cortical connectivity in EEG recordings following the administration of psychedelics, we aimed to extend our studies by using extracellular electrophysiological recordings[20,21]. In particular, we performed simultaneous dual Neuropixel recordings in MOs (PFC) and SSp (SM) of Ctrl, D1R-Nurr1cKO, and AAV-Nurr1cKO mice after the administration of DOI (1 mg/kg) (Fig. 7m–o). These electrophysiological recordings revealed a significant decrease in the unit firing rate correlations between PFC and SM of Ctrl brains (Fig. 7p, q). Nevertheless, this effect was significantly reversed in both D1R-Nurr1cKO and AAV-Nurr1cKO brains (Fig. 7p, q). Interestingly, the unit firing rate correlations within the same region did not show any significant difference between the groups (Supplementary Fig. 11).

Therefore, we provide evidence that the effects in intercortical correlation are dependent on Nurr1-induced expression of hallucinogen receptors in CLA.

## Discussion

There is a growing interest in CLA's role in the brain. Rodent CLA ontogenesis is characterized by the onset of strong Nurr1 expression at prenatal stages[15,16]. Therefore, investigation of Nurr1's involvement in CLA's activity would potentially unveil crucial aspects about the function of this elusive brain structure. We highlight that Nurr1 is crucial in regulating the transcriptional identity of CLA in a manner which is different from the function of this transcription factor in the ventral midbrain. Disruption of Nurr1's expression during prenatal stages causes agenesis of midbrain DA neurons[17,31]. The absence of DA neurons is confirmed by the loss of retrograde labeling of nigrostriatal neurons but also the ablation of DA cell-related genes[17,31]. Conversely, we demonstrate that CLA Nurr1 deletion in both early and adult life stages produces a robust reduction of CLA-enriched genes without causing CLA neuron agenesis or loss of axonal projections of CLA neurons. Hence, the current study is fundamental for the future investigation the Nurr1 as a key transcription factor for CLA neuron molecular characteristics. A question that arises from this study is how Nurr1 activity contributes to the specification of the CLA. Nurr1 is known for its ability to form homodimer and heterodimer complexes that bind to distinct transcription regulatory elements[44,45]. Additionally, Nurr1's post-translational modifications add another layer of complexity to its function as a transcription factor[44]. These modifications enable Nurr1 to navigate different hubs within the transcription network hierarchy, depending on the brain region's expression profile. While this study provides evidence that Nurr1 in the CLA holds a prominent position in the transcription network hierarchy, the molecular mechanism underlying this phenomenon remains to be clarified in future research.

It has been reported that CLA neurons belong to the same transcriptomic cell type, but recent studies describe several CLA subtypes with distinct morphological and electrophysiological features[7]. To pinpoint the role of Nurr1 in various CLA neuron types, an intersectional transgenic approach with highly selective labeling of specific CLA subpopulations could be interesting. For instance, there is possibility that loss of Nurr1 may differentially alter the electrophysiological properties or the axonal arborization span in the cortex of the discrete CLA neuron subgroups. Since CLA-related research is rapidly expanding[32], our study provides valuable insights about the regulation of the transcriptional identity of the structure.

CLA's role in various behaviors has gained much attention from the neuroscientific community[32]. Even though several studies attempt to delineate the behavioral readouts that occur after CLA neuron

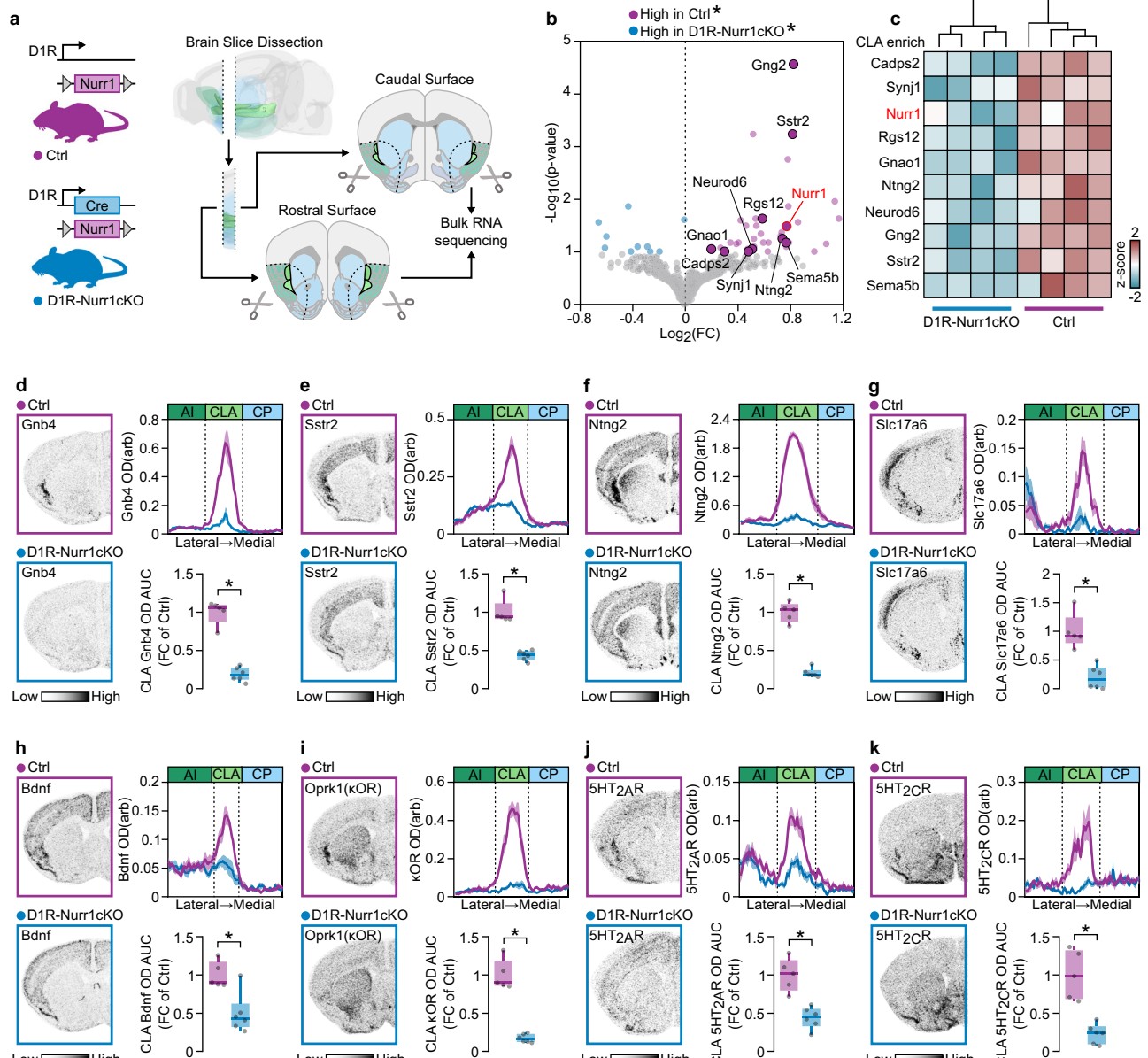

**Fig. 5 | D1R-Nurr1cKO mice display reduced expression of CLA-enriched genes. a** Illustration of D1R-Nurr1cKO mouse strain construct (left) and the coronal brain levels that were dissected for bulk RNA sequencing (right). **b** Volcano plot showing the CLA-enriched genes that were significantly downregulated in D1R-Nurr1cKO mice (Ctrl: *n* = 4, D1R-Nurr1cKO: *n* = 4; *p*-value adjusted < 0.1, Benjamini & Hochberg correction). **c** Heatmap showing the expression z-score of the CLA-enriched genes across the different samples. **d**–**k** ISH autoradiographs (left) and graphs (right), showing Gnb4 (**d**), Sstr2 (**e**), Ntng2 (**f**), Slc17a6 (**g**), Bdnf (**h**), Oprk1 (κOR) (**i**), 5HT₂AR (**j**) and 5HT₂CR (**k**) mRNA at CLA of Ctrl and D1R-Nurr1cKO mice (Ctrl: *n* = 5,

D1R-Nurr1cKO: *n* = 6; **d**: **p* < 0.0001, **e**: **p* < 0.0001, **f**: **p* < 0.0001, **g**: *p* = 0.0006, **h**: **p* = 0.0046, **i**: **p* < 0.0001, **j**: **p* = 0.0006, **k**: **p* = 0.0004, Two-sided Unpaired *t*-test). Data in line-graphs are expressed as mean ± SEM. Boxplots show all data points, the 25th and 75th percentile (box), the median (center), and the maxima (whiskers). OD: optic density, arb: arbitrary units, AI: agranular insula, CLA: claustrum/dorsal endopiriform cortex complex, CP: caudoputamen, AUC: area under the curve, FC: fold change, ISH: in situ hybridization. Source data are provided as a Source Data file.

silencing, CLA's function remains elusive[46]. Nevertheless, these inhibition studies support that CLA is involved in multiple behavioral tasks such as attention, pain processing, anxiety, and depression-like behaviors[5,33–35]. Our study's main scope is limited to Nurr1 effects, and we show that even though Nurr1 deletion diminishes many CLA-specific genes, it does not affect performance in specific behavioral tasks. There is a possibility that Nurr1 does not directly inhibit CLA's function but rather limits its modulation due to the massive transcriptional changes that take place after Nurr1 deletion. Future studies could examine CLA-specific inhibition with Nurr1 ablation in parallel and assess the same behaviors that have been described to be affected

by CLA inhibition. Several of the studies that address the behavioral readouts of CLA inhibition employ transgenic mice that provide precise viral delivery to the structure[6,35,47]. One of the most CLA-specific transgenic approaches requires using mice expressing Cre under *Gnb4* promoter[7,47,48]. Interestingly, *Gnb4* is one of the top downregulated genes in our study, implying that Nurr1 deletion might interfere with the genetic targeting of CLA cells in this CLA-specific mouse line. While our study shows that Nurr1-ablated mice do not display deficit in any behavioral tasks described to be affected by CLA inhibition, we postulate insights about structure-specific manipulation that might be useful for future studies.

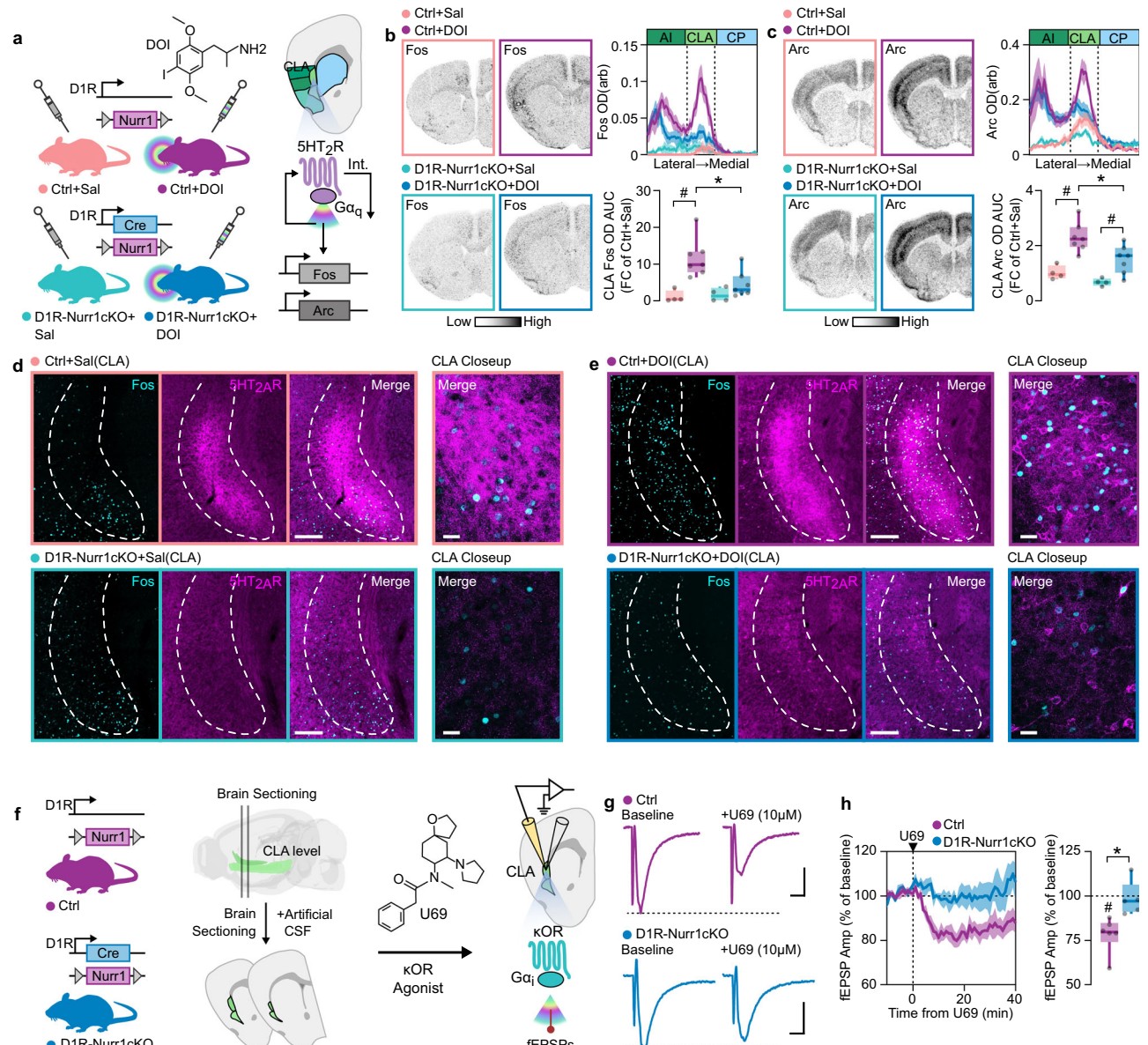

**Fig. 6 | D1R-Nurr1cKO display suppressed hallucinogen-receptor-induced effects in CLA. a** Illustration showing the DOI (8 mg/kg ip) administration experimental setup and the 5HT$_2$R-mediated effects on the CLA, (upregulation of Fos and Arc, 5HT$_2$R internalization). **b** ISH autoradiographs (Left) and graphs (Right) showing Fos mRNA in CLA of Ctrl and D1R-Nurr1cKO mice treated with DOI. (Ctrl +Sal: $n = 4$, Ctrl+DOI: $n = 7$, D1R-Nurr1cKO+Sal: $n = 4$, D1R-Nurr1cKO+DOI: $n = 7$; Two-way ANOVA, Genotype×Treatment: F(1,18) = 5.281, $p = 0.0338$; Ctrl vs D1R-Nurr1cKO: *$p = 0.0045$, Sal vs DOI: #$p = 0.0006$, Sidak's post-hoc test). **c** ISH autoradiographs (Left) and graphs (Right) showing Arc mRNA in CLA of Ctrl and D1R-Nurr1cKO mice treated with DOI. (Ctrl+Sal: $n = 4$, Ctrl+DOI: $n = 7$, D1R-Nurr1cKO +Sal: $n = 4$, D1R-Nurr1cKO+DOI: $n = 7$; Two-way ANOVA, Genotype: F(1,18) = 8.667, $p = 0.0087$, Treatment: F(1,18) = 29.42, $p < 0.0001$; Ctrl vs D1R-Nurr1cKO: *$p = 0.0061$, Sal vs DOI: #$p < 0.05$, Sidak's post-hoc test). **d, e** Fluorescent images showing the immunolabeling of 5HT$_2$AR and Fos in CLA of Ctrl and D1R-Nurr1cKO

mice treated with Sal (**d**) or DOI (**e**). **f** Illustration depicting the experimental procedure for recording fEPSPs following the application of the κOR agonist U69 (10 μM) in CLA sections, along with the proposed κOR-induced suppression of fEPSPs. **g** Representative current traces of extracellular fEPSP recording in CLA of Ctrl and D1-Nurr1cKO sections by using U69 (vertical scale-bar: 2 mV, horizontal scale-bar: 60 ms). **h** Line-graphs(left) and boxplots (right) showing the CLA fEPSP response of Ctrl and D1R-Nurr1cKO mice, after the application of U69. (*$p = 0.008$, Two-sided Unpaired *t*-test; #$p < 0.05$, Two-sided One-sample *t*-test, mean = 100). Data in line-graphs are expressed as mean ± SEM. Boxplots show all data points, the 25th and 75th percentile (box), the median (center), and the maxima (whiskers). DOI: 2,5-dimethoxy-4-iodoamphetamine, ip: intraperitoneal, CLA: claustrum/dorsal endopiriform cortex complex, Sal: saline, Int.: internalization, arb: arbitrary units, AUC: area under the curve, FC: fold change, U69: U-69,593, fEPSPs: field excitatory postsynaptic potentials. Source data are provided as a Source Data file.

Even though there is evidence on the molecular targets of hallucinogens, the exact brain structures that mediate the hallucinatory experience remain elusive. The default mode network (DMN) is mainly composed by medial PFC and posterior cingulate, while it has been proposed to exhibit the center of the 'ego'[49]. This self-referential processing led scientists to propose that decreased DMN activity can lead to ego dissolution caused by some hallucinogens[49]. Particularly, it is

observed that hallucinogens induce decreased cerebral blood flow in PFC, increased BOLD-fc of DMN with SM regions, and decreased global cortical synchrony in EEG[22,23,25,43]. Herein, we report that pfUS is a sensitive technology that can detect such changes in mice. Since 5HT$_2$R and κOR agonists do not display common behavioral effects in mice[50], our data in pfUS bridge the gap in the mechanism of action between psychedelics and opioidergic atypical hallucinogens. We

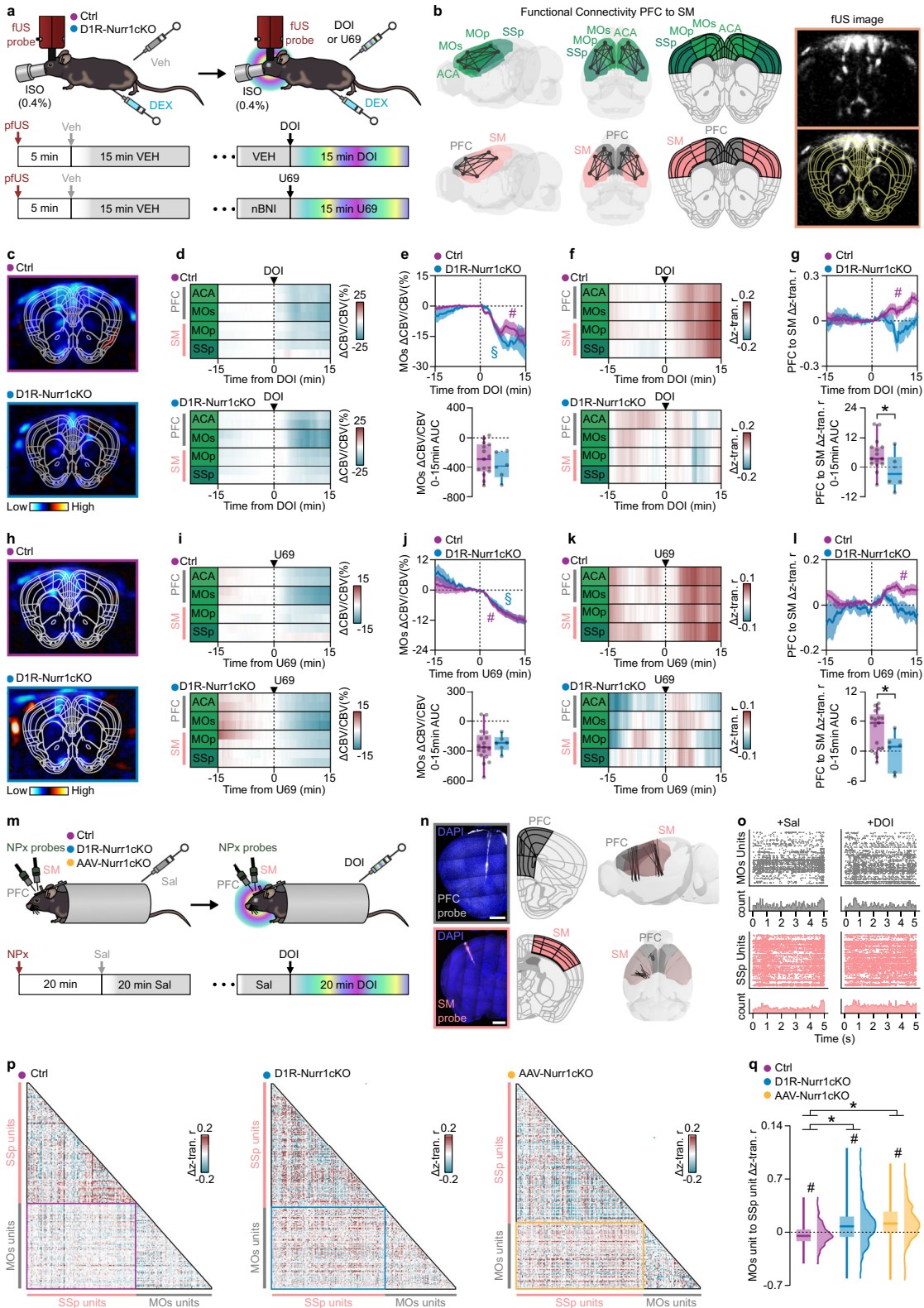

show that pfUS can be a useful tool to examine the hallucinogenic properties of certain compounds[51,52] and identify the brain region that is responsible for these effects. Due to its strong connections with multiple cortical regions, CLA has been hypothesized as the core brain structure that mediates delusions and hallucinations[4,9,49]. Considering its high density of κOR and 5HTRs, CLA appears to be a suitable neurobiological center of hallucinogens[9–11]. While others have suggested

the necessity of the claustrum for hallucinations[9–11,32,49], it has not been experimentally confirmed. Since CLA Nurr1 deletion abolished both the expression and the effects of hallucinogen receptors in the area, our mouse model is suitable for testing this hypothesis. This study indicates that loss of Nurr1 in CLA decreased the ability of 5HT$_2$R and κOR agonists to increase pfUS-fc. Additionally, the absence of Nurr1 in CLA abolished the 5HT$_2$R-induced reduction in electrophysiological

**Fig. 7 | Changes in cortical functional connectivity in response to hallucinogen receptor agonists are influenced by Nurr1 deletion in the CLA. a, b** Illustrations and coronal fUS image showing the experimental procedure the brain regions that were selected (PFC: gray, SM: pink). **c–e** pfUS images (**c**), heatmaps (**d**), and graphs (**e**) showing CBV changes after DOI ($p = 0.4181$, Two-sided Unpaired $t$-test). **f** Heatmaps showing the mean $\Delta z$-transformed-$r$-coefficient (correlation with the rest cortical regions) over time after DOI. **g** Graphs showing the $\Delta z$-transformed $r$-coefficient over time between PFC and SM (*$p = 0.045$, Two-sided Unpaired $t$-test). **h–j** pfUS images (**h**), heatmaps (**i**), and graphs (**j**) showing CBV changes after U69 ($p = 0.7830$, Two-sided Unpaired $t$-test). **k** Heatmaps showing the mean $\Delta z$-transformed $r$-coefficient (correlation with the rest cortical regions) change over time after U69. **l** Graphs showing the $\Delta z$-transformed $r$-coefficient change over time between PFC and SM regions (*$p = 0.0282$, Two-sided Unpaired $t$-test). **m** Illustration showing the neuropixel probe electrophysiological recordings. **n** Fluorescent images and 3d brain reconstruction showing the course of neuro-pixel probes in the brain (scale-bars: 1 mm). **o** Represenative raster plots showing the firing rate of the units after DOI. **p** Representative correlograms showing the $\Delta z$-transformed $r$-coefficient between the units of the two neuropixel probes (MOs and SSp probe). **q** Boxplot/violin plot graph showing the MOs to SSp unit $\Delta z$-transformed $r$-coefficient (Ctrl mice: $n = 3$, Ctrl-units: $n = 281$, D1R-Nurr1cKO-mice: $n = 4$, D1R-Nurr1cKO-units: $n = 387$, AAV-Nurr1cKO-mice: $n = 3$, AAV-Nurr1cKO-units: $n = 351$; #$p < 0.001$, Two-sided One-sample Wilcoxon signed rank test, mean = 0; Kruskal–Wallis test, $p < 0.0001$, *$p < 0.01$, Dunn's post-hoc test). Data in line-graphs are expressed as mean ± SEM (Ctrl: DOI/U69 vs baseline: #FDR < 0.05, D1R-Nurr1cKO: DOI/U69 vs baseline: §FDR < 0.05). Animal numbers in all pfUS graphs: Ctrl: $n = 15$, D1R-Nurr1cKO: $n = 6$. Boxplots show all data points, the 25th and 75th percentile (box), the median (center), and the maxima (whiskers). pfUS: pharmacological functional ultrasound, fc: functional connectivity, DOI: 2,5-dimethoxy-4-iodoamphetamine, U69: U-69,593, ISO: isoflurane, DEX: dexmedetomidine, Veh: vehicle, SSp: primary somatosensory area, ACA: anterior cingulate area, MOs: secondary motor area, MOp: primary motor area, PFC: prefrontal cortex, SM: sensorimotor areas, CBV: cerebral blood volume, AUC: area under the curve, NPx: neuropixel, Sal: saline. Source data are provided as a Source Data file.

brain coherence. Therefore, we demonstrate that CLA might be the structure that mediates the connectivity changes observed in hallucinogenic-like states. This coincides with the theory that the CLA is responsible for modulating the activity of spatially distant PFC and SM cortical areas[32]. However, 5HT$_2$R and κOR agonists' effects on PFC CBV were not affected by the CLA Nurr1 deletion. At the same time, the loss of Nurr1 in CLA does not abolish the HTR, which is the most widely used behavioral tool of measuring psychedelic effects on mice[50]. Even though there are non-hallucinogens that elicit HTR in rodents, it remains the golden standard for the assessment of hallucinatory effects. These results indicate that CLA might play a concerted role in hallucinatory experience rather than being the principal brain area that mediates the hallucinations[53]. While hallucinogens produce multiple brain changes, which are probably facilitated by different brain areas, we provide evidence that the CLA represents a point of convergence for hallucinogenic serotoninergic and opioidergic receptors. Moreover, the current study may pave the road for the design of new experimental strategies to investigate the functional role of CLA in the brain.

It is important to note that the current study has potential limitations. For instance, both of our Nurr1-ablation strategies might lead to off-target deletion of the transcription factor in other brain regions, which could contribute to our results. Nurr1 expression is dynamic and has been reported to upregulate under certain conditions in specific brain areas, such as the hippocampus[54]. Considering this, disabling Nurr1 transcription in brain areas that do not typically display strong Nurr1 expression may impair the ability of these cell types to increase the transcription factor's mRNA and subsequent protein levels. Another limitation arises from using pfUS to assess the effects of hallucinogens in brain regions. While it is true that neurovascular coupling causes changes in CBV due to increased neuronal activation during tasks, this is not always the case[55]. In pharmacological studies, the situation is more complex, as hallucinogens may have direct, nonspecific vascular effects unrelated to neural activity. This is supported by evidence that 5HT$_{2A}$Rs can directly affect blood flow[56]. Acknowledging these limitations is crucial for interpreting our findings accurately.

The discovery that Nurr1 serves as a major regulator for the gene transcription profile of CLA lays the groundwork for the elucidation of CLA functions. In particular, by understanding the relationship between CLA's neuromodulation and hallucinations, we may gain crucial insights into how the CLA's function can be hijacked in psychosis and other psychiatric diseases.

## Methods
### Animals (behavioral experiments)
Mice with C57BL6 genetic background (2–6 months old male and female) were housed in air-conditioned rooms (12-h dark/light cycle) at 20 °C and a humidity of 53%, with free access to food and water, provided ad libitum. Sprague-Dawley rats (2–6 months male and female) that were used in radioactive ISH experiments of Fig. 1, were housed in the same conditions as the mice. Nurr1-heterozygous (+/−) mouse construct was designed and generated by Rojas et al.[27]. Our Nurr1-flx mouse line was generated by the insertion of two loxP sequences in the second and third introns, so that the coding sequence of the first coding exon is excised at the presence of Cre[31]. This mouse line was crossed with mice expressing Cre under the D1R promoter. As conditional KO mice for each Cre-line were considered those which were heterozygous for Cre (+/−) and homozygous for Nurr1-floxed (+/+). The transgenic D1R-Cre (EY262) mouse line has been obtained from GENSAT. For the evaluation of Cre expression, D1R-Cre mice were crossed with a floxed mCherry mouse line and the combined heterozygous D1R-Cre (+/−) and floxed mCherry (+/−) offsprings were used for imaging. In this study, apart from the pfUS section, the Ctrl mice used were Nurr1-flx mice. For pfUS experiments with DOI or U69, the Ctrl mice included both Nurr1-flx and WT mice (Cre: −/−, floxed Nurr1: −/−). WT mice were also used in retrograde tracing with rgAAV-Cre and in pfUS experiments that included pretreatment with KET or nBNI. Housing and experimental procedures were fully approved by the local ethical committee at Karolinska Institute and Animal Ethics Board (N1525-2017,3218-2022 ;1535-2024) and conducted in accordance with the European Communities Council Directive of 24 November 1986 (86/609/EEC). Throughout all experimental procedures, we ensured the use of an equal number of female and male mice. Mice whose brains were not used for further experiments were euthanized using carbon dioxide chambers. The behavioral experiments that were performed on the animals are detailed in Supplementary information.

### Stereotactic injections and viruses
Mice were anesthetized with 1.5%–2.5% isoflurane and injected with 1 µl (0.2 µl/min) of Fluorogold (Fg) (Fluorochrome) containing solution (1–2% in saline) or AAVs. For conditional Nurr1 deletion, we used 4 µl (1 µl in each injection site) of a GFP-Cre expressing AAV5 under the human synapsin promoter (AAV5-hsyn-GFP-Cre for ST experiments and AAV5-CamKIIa-GFP-Cre for neuropixel recordings; UNC Vector Core). As a control virus, we used a solely eGFP expressing AAV5 under the same promoter (AAV5-hsyn-eGFP; UNC Vector Core). For the conditional retrograde Nurr1 deletion, we used 300 nl of a rgAAV-Cre (rgAAV-Ef1a-Cre; Addgene). For the labeling of CLA cells infected retrogradely from rgAAV-Cre, we used 300 nl of a Cre-dependent AAV5-DIO-GFP (AAV5-EF1a-DIO-eYFP; UNC Vector Core). The number of retrogradely labeled cells in this experiment was determined from three representative sections. The coordinates for the injections in mPFC were AP: +2.2 mm, ML: ±0.4 mm, DV: −1.4 mm relative to bregma

and dural surface. Mice injected with Fg were sacrificed 7 days after the surgery. The coordinates for rostral and caudal AAV injections in CLA were AP: +1 mm, ML: ±2.8 mm, DV: −2.4 mm and AP: 0 mm, ML: ±3.5 mm, DV: −2.6 mm relative to bregma and dural surface, respectively. Mice injected with AAVs were sacrificed 8–10 weeks after injection.

## Marmoset and human brain tissue

Marmoset (*Callithryx jacchus*) fresh frozen brain sections used in the papers were obtained from previously published work[57]. The human brain section that was used for radioactive ISH was acquired from the University College of London, Queens Square Brain Bank. In detail, the current fresh frozen brain section was obtained from an 88-year-old male (p31/11). The current human brain sample used in this study was obtained following informed consent from the donor or the donor's relatives. The study was approved by the appropriate ethical review board, with the ethical approval number 2014/1366-31.

## Animals (behavioral experiments)

Y-maze: Mice were placed at the center of a symmetric Y-maze with arms measuring 40 cm long, 8 cm wide, and 20 cm high and recorded with an overhead camera for 10 min. An alteration is counted when the mouse entered the 3 different arms during a triad on overlapping triplet sets (e.g., in the sequence ACBCBACBAA, four alternations were counted). The spontaneous alternation was calculated by counting the number alternations as percentage of the total number of arm entries minus 2.

Passive avoidance test (PAT): Mice were placed in the light compartment of a step-through passive avoidance apparatus (Ugo Basile) for 60 s, before a sliding door was opened and the mouse entered a dark compartment (training latency). After the animal stepped into the dark compartment with all four paws, the door automatically closed and a weak electrical stimulus (0.3 mA, 2 s duration, scrambled current) was delivered through the grid floor. Immediately post-shock, mouse vocalizations were recorded for 5 s and the number of vocalizations emitted as well as vocal amplitude were measured using Avisoft Ultrasound Gate Analyzer (Avisoft Bioacoustics). After a 24-h delay, a mouse was again placed in the light compartment and the step-through latency to return to the dark compartment was measured (retention latency).

Feeding with auditory destructor: The current behavioral test was designed as a modified version of pup-retrieval test from Atlan et al. 2018, by adapting a novelty-induced hypophagia protocol used in mice56. Mice were single-caged and habituated 10 min to condensed milk (1:3 diluted in tap water) for three days. On the 4th day the milk consumption and the latency to feed were measured. The 5th day the milk consumption and the latency to feed were measured again with the presence of a loud auditory destructor (song Pluto by the artist Bjork). The performance of the mice was calculated as the delta between the 5th and the 4th day.

Pre-pulse inhibition test (PPI): For pre-pulse inhibition testing we used two startle chambers (San Diego Instruments, San Diego, Calif., USA). Each chamber contained a Plexiglas cylinder attached to a platform and a loudspeaker that produced both a continuous background noise of 65 dB and different acoustic stimuli. Mouse startle responses caused vibrations of the cylinder, which were transformed to analog signals by a piezoelectric transducer under the platform. Calibrations were performed on the chambers before every experiment. Each test session started with a 5-min background noise (65 dB white noise) habituation period. The background noise continued throughout the test session. The habituation period was followed by four blocks of trials, with an average interval of 12 s. The first and the sixth block consisted of five Startle (a 40-ms 120 dB burst) trials. Four trial types were presented during the second and the third block in a pseudo-randomized sequence with ten of each trial type per block.

Trial types contained a Startle and three separate pre-pulse trials in which 20-ms long prepulses of different intensities (68, 71, or 77 dB) preceded the startle stimulus by a 100 ms interval. The test session lasted for a total of 23 min and contained 60 trials.

Forced swim test (FST): Mice were allowed to habituate in the behavioral room 30 min before the test. Afterward, the mice were individually placed in cylindrical tanks (diameter x height: 20 × 50 cm) filled with water to a depth of 30 cm (water temperature: 25 ± 1 °C). Over a test duration of 7 min, the behavior was recorded and analyzed via an automated video tracking system (EthoVision XT 8; Noldus).

Sucrose preference test (SPT): Mice were single-housed and given a choice between two bottles containing either tap water or a 1% sucrose solution in their home cages for 48 h. The position of the two bottles was switched after 24 h to avoid a side preference. Water and sucrose consumption were measured each day at the end of the afternoon by weighing the bottles. Sucrose preference (sucrose solution consumption (g)/water consumption (g) + sucrose solution consumption (g)) was calculated over the 48 h-period.

Head twitch response (HTR): Mice were treated with saline or with DOI (1 mg/kg or 8 mg/kg, ip; Sigma), immediately placed in an empty cage. Over a test duration of 10 min, the behavior was recorded and analyzed via a video camera that was placed above the cage. All the mice of the AAV experiments were first treated with saline and recorded for 10 min. Subsequently, the same mice were treated with DOI (8 mg/kg, ip) and further recorded for 10 min. An HTR was defined as a clear, rapid, oscillatory head movement of the mouse. At least two observers counted HTRs in a 10-min period.

Open field test (OFT): Mice were placed in a 55 cm × 55 cm squared box for measurement of locomotion during a trial duration of 20 min. For the subchronic cocaine pharmacological studies, mice were treated daily for 6 days with intraperitoneal (ip) saline or cocaine (10 mg/kg) prior testing. The 7th-day mice were treated with ip saline or cocaine right before the OFT and were recorded for 30 min. For the acute pharmacological studies, mice were treated with ip saline right before the test and were recorded for 60 min. The same mice were later (the same or the day after) administered amphetamine (5 mg/kg, ip) or PCP (8 mg/kg, ip; Sigma) solutions (dissolved in saline), placed in open field arena, and recorded for another 60 min. Total distance traveled was measured using an automated video tracking system (EthoVision XT 8; Noldus).

## Spatial transcriptomics - visium spatial gene expression technology and sequencing

Coronal sections were taken at the claustrum level for one hemisphere of each mouse brain. Fresh frozen tissue sections were 10-μm thick and placed on Visium slides (10x Genomics, PN: 2000233), which were subsequentely stored at −80 °C for less than a week before proceeding with the 10x Genomics Visium Spatial Gene Expression protocol (User Guide, CG000239 Rev F) following the manufacturer's instructions. The staining time was optimized to 5 min for hematoxylin and 45 s for eosin. Brightfield images were acquired using a Zeiss AxioImager.Z2 VSlide microscope that uses the Metasystems VSlide scanning system with Metafer 5 v3.14.3 and VSlide software. The permeabilization time was optimized to 14 min. In total 6 Visium libraries were prepared and amplified with 15 PCR cycles. Samples were indexed using the 10X Genomics Dual Index Kit TT, Set A (PN = 1000215). Final Visium libraries were sequenced using an Illumina NextSeq 2000 instrument and P3 200 cycle kit (PN: 20040560). Read1 and Read2 were 28 and 150 bp long, respectively.

## Spatial transcriptomics - counts data generation and processing

The counts matrices from the corresponding fastq sequences were processed using Spaceranger count (v1.3.1, 10x Genomics Inc.). The counts matrices were then analyzed using the R package Seurat (v4.1.1), Harmony (v0.1.0), and SCTransform (v0.3.4).

## Spatial transcriptomics – data analysis

Individual sections were assessed for overall quality using spatial plots showing UMI distribution and Genes distribution per spot across the whole tissue section. The whole dataset was filtered for mitochondrial genes, ribosomal genes, hemoglobin genes, and genes with 0 UMIs detected. Additional spot-level filtering was applied to only include spots with UMI count above 100 and detected genes count above 200. Individual sections were assessed again after filtering using spatial heatmaps and violin plots. Individual count matrices were then normalized using SCTransform (method = "glmGamPoi"). The normalized count matrices were integrated using integration anchors following an established Seurat package based on visium data analysis workflow. Dimensionality reduction was performed using PCA and UMAP for 50 dimensions and the first 40 dimensions were later used for further analysis after assessment using ElbowPlot() and DimHeatmap(). Batch effects caused by handling of individual tissue sections were removed using RunHarmony() (group.by.vars = "sample_id" and theta = 2). Clustering was performed using FindClusters() with the original Louvain algorithm and at resolution 2. The final clusters were then annotated using cluster marker genes generated by FindAllMarkers(). Visium spots falling under the CLA region were manually selected in Loupe Browser (v6.3.0, 10x Genomics Inc.) using prior knowledge of the mouse brain tissue anatomy. Differential expression (DE) analysis was then performed for these manually picked spots between the AAV-Nurr1cKO sections and the AAV-Ctrl sections. Two-sided Wilcoxon's rank sum test was performed for differential analysis with multiple-testing adjusted P-values computed by PRESTO. adjusted P-values estimates of less than 0.1 were later applied as a threshold[58].

## Bulk RNA-sequencing

Libraries from 8 samples (4 Ctrl and 4 D1R-Nurr1cKO) for bulk RNA-sequencing were generated using the Smartseq2 protocol[59]. Smartseq2 for sensitive full-length transcriptome profiling in single cells[59]. cDNA libraries were tagmented using home-made Tn5 enzyme[60]. Tn5 transposase and tagmentation procedures for massively scaled sequencing projects[61] and Nextera dual indexes (Illumina). The quality of cDNA and tagmented cDNA was checked on high-sensitivity DNA chip (Agilent Bioanalyzer). The reference transcriptome Mus_musculus.GRCm39.cdna.all (release-107) from Ensembl was used. Salmon/1.6.0 was used to create an index on the transcriptome which is used as a structure for quasi-mapping of RNA-seq reads during alignment and quantification[62]. Salmon Index was made with parameters such as the –gcBias flag to correct for the common systematic biases in RNA-seq data, making it possible correct for fragment-level GC biases in the data. The other parameter was the –validateMappings flag for selective alignment. Next, "tximport" package version 1.20.0, was used to import and summarize the transcript abundances and construct of a transcript-level DESeqDataSet object from Salmon output files. This package was also used to construct a tx2gene table for linking transcripts to genes using Gencode Mouse Transcript Annotation (GTF file) Release M30 (GRCm39). Next, the necessary quantification data for DESeq2 was imported using the tximport function. Finally, DESeq2 package version 1.34.0 was used to construct a DESeqDataSet from the txi object, using sample information and design = ~condition[63]. Prefiltering was done by removing transcripts with less than 3 reads across all 8 samples. Differential gene expression analysis (Ctrl vs D1R-Nurr1cKO) was done with results() function in the DESeq2 package, using Wald statistical test and p-value adjustment with Benjamini & Hochberg (FDR < 0.1) correction. 'rtracklayer' package version 1.52.1 was used to extract "gene_id", "gene_name", and "transcript_id" from the Gencode annotation GTF file to later match them to the (DE) results gene IDs. lfcShrink() function in DESeq2 package was used to estimate the log fold changes (LFCs) and shrink them toward zero to avoid the small counts values' otherwise inaccurately estimated large LFC, from affecting and dominating the real high-ranked LFCs. The adaptive t prior shrinkage estimator type = "apeglm" was used from the apeglm package[64]. The 'EnhancedVolcano' package version 1.10.0 and the 'pheatmap' package version 1.0.12 were used to make volcano plots and the heatmap.

## Slice electrophysiology

Mice underwent cervical dislocation followed by decapitation. The brains were rapidly removed and submerged in a slicing solution containing glucose (10 mM), $NaH_2PO_4$ (1 mM), KCl (2 mM), NaCl (15.9 mM), sucrose (219.7 mM), $MgCl_2$ (5.2 mM), CaCl2 (1.1 mM), and $NaHCO_3$ (26 mM). Coronal sections (400 μm thick) containing claustrum were obtained using a microslicer (Leica). The sections were incubated in a modified artificial cerebrospinal fluid (aCSF) containing glucose (10 mM), $NaH_2PO4$ (1.2 mM), $NaHCO_3$ (23.4 mM), KCl (2.5 mM), NaCl (126 mM), $MgCl_2$ (4.7 mM), and $CaCl_2$ (1 mM) at 32 °C for 1 h after the slicing and then at 28 °C. We used a glass electrode filled with aCSF, positioned on the slice surface in the CLA, to record extracellular field potentials. In detail, the electrode was positioned adjacent lateral to the external capsule. We used a stimulating electrode (concentric bipolar, FHC) placed in the slice, near the recording electrode, to evoke fEPSPs. Stimulation pulses were applied every 15 s to the brain slice through the stimulation electrode. Single stimuli were applied at an intensity yielding 50–60% maximal response as assessed by a stimulus/response curve established, for each slice, at the beginning of the recording session, by increasing the stimulation intensity and measuring the amplitude of the fEPSP. Signals were amplified via a GeneClamp 500B amplifier (Axon Instruments), acquired at 10 kHz, and filtered at 2 kHz. We used the pClamp 10 software (Axon Instruments) to acquire and analyze data. U-69,593 (Sigma) stock solution was prepared in 0.1 M HCl and further diluted in aCSF at 10 μM final concentration.

## Pharmacological functional ultrasound (pfUS)

The animals were sedated with 4% isoflurane and maintained at 1.5% upon fixation in a stereotaxic frame (David Kopf Instruments). Animal core temperature was maintained between 36 and 37 °C with a homeothermic warming system (PhysioSuite, Kent Scientific Corporation). To prevent discomfort from stereotaxic fixation mice were given Meloxicam (Metacam®, Boehringer Ingelheim) subcutaneously at 0.05 mg/kg. Hair on the scalp was removed via electrical shaving. A subcutaneous bolus injection of 0.067 mg/kg dexmedetomidine hydrochloride (Tocris) in physiological saline was given, directly followed by continuous infusion at 0.2 mg/kg/h, 5 mL/kg flow. Five minutes into infusion, gradual isoflurane turn-down started at 0.2% per minute until 0.4%. Functional imaging was initiated no earlier than 10 min after the isoflurane rate at 0.4%. Doppler vascular images were obtained using the Ultrafast Compound Doppler Imaging technique[65]. Images were acquired for 40 min at a 1000 Hz frame rate. A fast scan of 6 mm volume with successive images taken on multiple coronal planes was performed for positioning the probe in the selected plane (approximately AP: +1.4 mm relative to bregma). The first injection with vehicle (saline or 10% Tween-80) was applied 5 min after acquisition start. 15 min after the vehicle injection, the mice were treated with DOI (1 mg/kg, ip) dissolved in saline or U69 (1 mg/kg, sc; Boehringer Ingelheim) dissolved in 10% Tween-80. Pretreatment involved administering KET (3 mg/kg, ip; Boehringer Ingelheim) 30 min before the vehicle injection, and nBNI (32 mg/kg, sc; Boehringer Ingelheim) 24 h before the vehicle injection. Dexmedetomidine sedation was antagonized with a subcutaneous injection of atipamezole (Alzane, Zoetis) at a dose of five times of total administered dexmedetomidine. Regional Power Doppler timecourses were extracted from the bilateral anterior cingulate, secondary motor, primary motor, and primary sensory cortices. The raw Power Doppler signals were despiked of artifacts using a protocol adapted from Brunner et al.[66]. After despiking, a band-pass filter ranging from 0.01 to 0.2 Hz was applied. To

quantify changes in regional CBVs, we used the despiked timecourses prior to filtering. We calculated the ΔCBV/CBV percentage by defining the final 5 min before the challenge as baseline. We tested for significant CBV changes in each region by performing pair-wise t-tests to baseline raw CBV values and controlled for multiple comparisons using the Benjamin–Hochberg FDR correction. To determine the effect of the pharmacological challenges on pfUS-fc we employed a sliding-window approach. Specifically, we generated regional Pearson's r correlation strengths by averaging all Fisher's z-transformed correlations of each bilateral region for every subject and time window. To detect pharmacological effects on pfUS-fc we defined the final 5-min window (15–20 min after the start of the scan) before the challenge as baseline and performed pair-wise t-tests for every subsequent edge and regional connectivity strength to their own respective values at baseline. We tested for significant pfUS-fc changes in each region by performing pair-wise t-tests to baseline and controlling for multiple comparisons using FDR correction.

### Radioactive in situ hybridization (ISH)

Mice underwent cervical dislocation followed by decapitation. The radioactive ISH was performed in cryostat (Leica) fresh frozen sections (12 μm thick). $^{35}$S-labeled anti-sense cRNA probes were prepared by in vitro transcription from cDNA clones corresponding to fragments of Arc, Fos, Nurr1, Gnb4, Ntng2, Sstr2, Bdnf, Slc17a6, Tmem163, D1R, κOR, 5-HT2AR and 5-HT2CR. The transcription was performed from 50–100 ng of linearized plasmid using [$^{35}$S] UTP (1250 Ci/mmol; Perkin Elmer) and SP6, T3, T7 RNA polymerase. Cryostat fresh frozen sections (12 μm thick) were post-fixed in 4% PFA for 5 min at room temperature, rinsed twice in 4 × sodium chloride– sodium citrate buffer (SSC), and placed into 0.25% acetic anhydride in 0.1 M triethanolamine/4 × SSC (pH = 8) for 10 min at room temperature. After dehydration in graded alcohols, the sections were hybridized overnight at 55 °C with 35S-labeled probe in 50 μl of hybridization solution (20 mM Tris– HCl/ 1 mM EDTA/300 mM NaCl/50% formamide/10% dextran sulfate/ 1 × Denhardt's/250 μg/ml yeast tRNA/100 μg/ml salmon sperm DNA/ 0.1% SDS/0.1% sodium thiosulfate). The slides were washed in 4 × SSC (5 min, four times), RNAse A (20 μg/ml) (20 min, at 37 °C), 2 × SSC (5 min, twice), 1 × SSC (5 min), 0.5 × SSC (5 min) at room temperature and rinsed in 0.1 × SSC at 65 °C (30 min, twice) (all washes contained 1 mM DTT), before being dehydrated in graded alcohols. The slides were then exposed on X-ray films for 4–15 days. For radioactive ISH studies of Fos and Arc, mice were sacrificed 1 h after DOI treatment. For CLA identification and OD quantification, we generated the average signal intensity traces within a rectangular region by using the "Plot profile" function in Fiji software. For these measurements, the rectangular area was placed in approximately the same location, centered in CLA in each section. CLA's center was determined as the transition point between cortex and striatum at the lateral end of the corpus callosum/ external capsule. The measurements from this trace, used to calculate the area under the curve, were defined as the high signal values around the rectangle center in the control brains.

### Fluorescent ISH (RNAscope)

Mice underwent cervical dislocation followed by decapitation. The Fluorescent ISH (RNAscope) was performed in cryostat (Leica) fresh frozen sections (12 μm thick) by using the RNAscope Multiplex Fluorescent Assay (Advanced Cell Diagnostics, cat. 320850). Cryostat (Leica) fresh frozen thaw-mounted sections (12 μm thickness) were post-fixed in 4% PFA for 15 min at 4 °C and dehydrated in graded alcohols. Then, it was applied Protease IV (Advanced Cell Diagnostics, cat. 322340) for 30 min at room temperature. Afterwards, the sections were hybridized with the probes: Nurr1 (Mm-Nr4a2, cat. 423358) and D1R (Mm-Drd1-C3, cat. 461901-C3) for 2 h at 40 °C. After the hybridization step, 4 standardized steps of amplification were followed (Amp 1-FL 30 min at 40 °C, Amp 2-FL 15 min at 40 °C, Amp 3-FL 30 min at

40 °C, Amp 4C-FL 15 min at 40 °C). Right after the last amplification step the sections were counterstained with DAPI for 30 s and covered with coverslips by using Dako fluorescent medium. The slides were imaged on an LSM 880 (Carl Zeiss) confocal microscope using a 63 × 1.4 NA oil immersion objective. Z-stacks of 6–9 μm thickness were obtained in each caption.

### Immunofluorescence

Mice were anesthetized and transcardially perfused with (4% paraformaldehyde). The frozen perfused brains were cryostat (Leica) cut in 30–40 μm free-floating sections and washed 2 times for 5 min with phosphate buffer solution (PBS). Then, they were incubated in blocking buffer (5% goat serum, 0.3% Triton-X in PBS) for 1 h. After this step they were applied with primary antibodies: anti-Nurr1 (cat. sc-990 (E-20); 1:50), rat anti-Fos (cat. 226 003; 1:200), rabbit anti-5HT$_{2A}$R (cat. RA24288; 1:200) and mouse anti-Cre (cat. MAB3120; 1:500) dissolved in blocking buffer and left overnight at 4 °C. The next day the slides washed 3 times for 5 min with PBS and incubated with fluorescent anti-rabbit, anti-mouse or anti-rat secondary antibodies solution for 2 h. After 3 times 5 min washing the sections were mounted on coated microscopy slides and covered with coverslips by using Dako fluorescent medium. The slides were imaged on an LSM 880 (Carl Zeiss) confocal microscope using a 20 × 0.8 NA objective. Z-stacks of 15–20 μm thickness were obtained in each caption. For immunofluorescent detection of Fos and 5HT$_{2A}$R internalization, mice were sacrificed 2 h after DOI treatment.

### Neuropixel recordings

The detailed description of the implant placement is described in Supplementary information. A total of 12 mice (4 Nurr1-cre, 4 Virus inject, and 4 C57BL/6 J) were injected sc with saline and DOI (1 mg/kg) while being head-fixed. The researcher who performed the recordings was blinded to the mice cohort. Mice were first habituated to be head-restrained and handled by the experimenter over a period of three to five days to reduce stress levels. The experimental protocol was divided into three 20 min blocks. Each block being separated by at least one minute from the others. Vehicle and DOI sc injections were performed respectively after block1 and block2. The behavior of the mice was monitored with a Blackfly camera (Teledyne FLIR, USA) during each block with a sampling frequency of 50 Hz.

### Neuropixel recordings (implant placement)

The mice were anesthetized with isoflurane (3% for induction then 1–2%). Buprenorphine (0.1 mg/kg sc), carprofen (5 mg/Kg sc) and lidocaine (4 mg/kg sc) was administered. The body temperature was maintained at 37 °C by a heating pad. An ocular ointment (Viscotears, Alcon) was applied over the eyes. The head of the mouse was fixed in a stereotaxic apparatus (Kopf). Lidocaine 2% was injected locally before the skin incision. The skin overlying the cortex was removed, the skull was cleaned with Chlorhexidine, and the bone gently cleaned. A thin layer of glue was applied on the exposed skull. A lightweight metal head-post was fixed with light-curing dental adhesive (OptiBond FL, Kerr) and cement (Tetric EvoFlow, Invoclar Vivadent, Schaan, Liechtenstein). For extracellular recordings, a chamber was made by building a wall with dental cement along the coronal suture and the front of the skull. Brain regions were targeted using stereotaxic coordinates. After the surgery, the animal is returned to its home cage, and carprofen (5 mg/Kg sc) was provided for postoperative pain relief 24 h following surgery. For acute Neuropixels recordings, two small craniotomies (300–500 μm in diameter) were opened a few hours (>3 h) before the experiment to access the pre-marked targeted cortical regions on the left hemisphere (PFC: +1.8 mm AP; 0.3 mm ML, SSp: −1.0 mm AP; 3.5 mm ML from Bregma). A gold pin was implanted over the right hemisphere to serve as a ground. The mice were anesthetized with isoflurane (3% for induction then 1–2%). Buprenorphine (0.1 mg/kg

sc), carprofen (5 mg/Kg sc) and lidocaine (4 mg/kg sc) was administered. The open craniotomy was covered with Silicone sealant (Kwik-Cast, WPI), and the mouse was returned to its home cage for recovery. Extracellular spikes were recorded using Neuropixels probes (Phase 3B Option 1, IMEC, Leuven, Belgium) with 383 recording sites along a single shank covering 3800 μm in depth. The probe was lowered gradually (speed ~20 μ.s$^{-1}$) in the left hemisphere with a micromanipulator (uMp-4, Sensapex, Oulu, Finland) until the tip reached a depth of ~3500–4200 μm under the surface of the pia. The probe was coated with CM-DiI (1,1′-dioctadecyl-3,3,3′3′-tetramethylindocarbocyanine perchlorate, Thermo Fisher, USA) a fixable lipophilic dye for post-hoc recovery of the recording location. The coating was achieved by holding a drop of CM-DiI at the end of a micropipette and repeatedly painting the probe shank with the drop, letting it dry, after which the probe appeared pink. The electrode reference was then connected to the ground pin, in contact with the PBS 1X solution filling the chamber. The probe was allowed to sit in the brain for 20–30 min before the recordings started. The signals were filtered between 0.3 Hz and 10 kHz and amplified. The data was digitized with a sampling frequency of 30 kHz with gain 500. The digitized signal was transferred to our data acquisition system (a PXIe acquisition module (PXI-Express chassis: PXIe-1071 and MXI-Express interface: PCIe-8381 and PXIe-8381), National Instruments), written to disk using SpikeGLX (Bill Karsh, Janelia) and stored on local server for future analysis.

### Neuropixel recordings (In vivo electrophysiological data analysis)

For probe tract reconstruction, mice were anesthetized and transcardially perfused with 4% paraformaldehyde. Brain sections were cut at a thickness of 400 μm, then cleared using the CUBIC protocol, and imaged at 4× magnification with a Zeiss 880 confocal microscope. The z-stacks containing the probe red fluorescent signal were subsequently pre-processed, registered to the Allen CCFv3, and the probe position was estimated using the DMC-BrainMap pipeline in napari (https://github.com/hejDMC/napari-dmc-brainmap). Unit locations were assigned based on the location of the electrode where that unit had the highest waveform amplitude. Three mice were excluded from the analysis: the first animal had no probe located in the PFC, the second had only 4 'good' units in SSp and the third had important electrical artifacts (amplifier saturation). For data analysis, spikes for each 'good' unit were binned in 50 ms intervals and smoothed by convolution with a Gaussian kernel (sigma = 2) and subsequently z-scored. For unit-fc analysis, we followed the approach used in Vesuna et al. 2020[67]. In detail, we calculated the unit-by-unit correlation (Pearson) for all MOs and SSp units was calculated, and the resulting values Fisher z transformed in 60 s windows slid by 30 s for the 15 min time period after vehicle/DOI injection per animal. Static fc matrices were calculated by averaging the correlation values per unit pair across time. Difference matrices (Fig. 7) were generated by subtracting the static fc matrices for the vehicle block from the DOI block. For quantification, average correlation values for each MOs unit to all SSp units was calculated (Fig. 7).

### Statistical analysis and graphs

Statistical analysis was performed by using Student's t-test, one-sample t-test, Mann–Whitney test, Kruskal–Wallis test, Two-way analysis of variance (ANOVA) or repeated measurements (RM) Two-way ANOVA using GraphPad Prism 7.0. Both Two-way ANOVA and RM Two-way ANOVA were followed by Sidak's post-hoc test. Kruskal–Wallis test was followed by Dunn's post-hoc test. All the graphs were made in GraphPad Prism 7.0. All the illustrations shown in the figures were made by using Inkscape 1.3.2. The 3D brain illustrations presented in the figures were created and obtained from the Scalable Brain Atlas[68–71].

### Reporting summary

Further information on research design is available in the Nature Portfolio Reporting Summary linked to this article.

## Data availability

The sequencing data generated in this study have been deposited in the NCBI-GEO database under accession code GSE229770. The corresponding count matrices, histology images, spotfiles from the CLA region, and final Seurat object from the ST data analysis are publicly available on the Mendeley Data project [72]. The GEO accession number for the bulk RNA-seq is GSE229732. Source data are provided with this paper.

## Code availability

Scripts required to reproduce the results and generated figures from the ST analysis are publicly available on our GitHub repository (https://github.com/giacomellolab/CLA-Nurr1)[73].

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

## Acknowledgements

Dr Yunting Yang for his help with genotyping the mice and maintain the colonies. I.M. was supported by a KI-NIH grant. P.S. was supported by grant from the Swedish Medical Research Council (2019-01422) and Knut and Alice Wallenberg Foundation. T.P., B.Y.S., and L.G. were supported by grants from the Swedish Medical Research Council (2020-0884) and Hjärnfonden (FO2023-0028)

## Author contributions

I.M., T.P., and P.S. conceived the study. I.M. and I.F. performed the animal experiments. I.M. performed the histological experiments. Y.M. and S.F. performed and analyzed spatial transcriptomics experiments. S.G. supervised Y.M. and S.F. T.I. performed the analysis of functional ultrasound experiments. M.S. designed and generated the probes for in situ hybridization experiments. K.T. and L.G. performed the bulk RNA-sequencing experiments. B.Y.S. performed the analysis of bulk RNA-sequencing data. X.Z. performed the electrophysiology experiments with support from KC, and supervised I.M. I.M. and P.L. performed the neuropixel recordings. P.L. and F.J. analyzed neuropixel recordings. M.C. supervised I.M., P.L., and F.J. in neuropixel experiments. B.H. supervised I.F. and T.I. I.M., B.H., T.P., and P.S. wrote the paper.

## Funding

## Competing interests

I.F. was employee at Boehringer Ingelheim Pharma GmbH & Co. KG during the time of the study. T.I. and B.H. are employees at Boehringer Ingelheim Pharma GmbH & Co. KG. S.F. and S.G. are scientific advisors to 10x Genomics Inc, which holds IP rights to the ST technology. S.G. holds 10x Genomics stocks. All other authors report no competing interests.
