## [Peer Review File · Nature Communications]

Claustrum/dorsal Endopiriform cortex complex cell-identity is determined by Nurr1 and regulates hallucinogenic-like states in miceREVIEWER COMMENTS

Reviewer #1 (Remarks to the Author):

Mantas et al. present a rather interesting study on the role of the transcription factor Nurr1 in regulating claustrum development and the ability for the claustrum to contribute to hallucinogenic states. They extensively show that conditional knock-out of the transcription factor Nurr1 alters gene expression in the claustrum including reduced expression of claustrum enriched genes. They further show that conditional Nurr1 knock-out reduces expression of receptors known to mediate hallucinogen-induced subjective experiences 5-HT_{2A}R, 5-HT_{2C}R, and KOR and confirm this finding functionally. Finally, they show that agonists of 5-HT_{2A}R and KOR increases functional connectivity between prefrontal cortex and sensorimotor cortices in a Nurr1-dependent manner. Several points require addressing to improve the manuscript.

Major

1. Nurr1 expression is apparent in the claustrum, but also in the ventrally-lying endopiriform nucleus as well as what looks like layer 6b neurons lining the lateral and dorsal aspects of the external capsule/corpus colosum. Receptor expression changes are not assessed in these areas to help address whether the fc findings are secondary to CLA receptor expression changes, or changes elsewhere. In addition, what independent verification exists to say that clusters 8 and 11 are the same anatomical region? 8 genetically looks like claustrum, but it could be that 11 is endopiriform nucleus. This needs to be addressed.
2. The authors use an indirect method of examining whether claustralcortical projections remain intact in the absence of Nurr1. Would suggest injecting rAAV-cre-gfp into a medial PFC of Nurr1-flx and wt mice and compare the number of retrogradely labeled neurons.
3. HTR is the gold standard for behavioral assessment of psychedelic (2a agonist) drug challenge in rodents. The lack of HTR changes following Nurr1 KO would strongly suggest that CLA 2a receptor has nothing to do with HTR, as the authors state, but also suggests that the CLA then has nothing to do with psychedelic states. This stands in conflict with the primary conclusions of this paper. Claiming that HTR is not a good readout because KOR agonists don't elicit them is not cogent, as KOR is a completely different receptor than 5HT_{2a} (unrelatedly, agonists to 2a are considered psychedelics and agonists of KOR are considered hallucinogens). The subjective responses to these drugs are vastly different, along vastly different timescales in humans.
4. AAV-Nurr1cKO was not used to bolster the D1R-Nurr1cKO fUS findings. This is a major weakness as indicated by the lack of HTR in the D1R-Nurr1cKO mice, which indicates that the "hallucinogenic" effects of DOI may lie outside the claustrum.
5. It is important to experimentally consider what would happen to CLA immediate early gene expression and fUS effects if a non-hallucinogenic drug were administered (eg meth, cocaine, ethanol or heroin). If nothing was found with these drug classes, this would strengthen the claim that the CLA is involved in hallucination, rather than non-specific drug responses writ large.
6. In figure 3 and 5, transcriptomics and in-situ hybridization are used to convincingly show that Nurr1 knock-out results in reduced expression of several genes enriched in the claustrum. However, the lack of a sufficiency experiment showing that Nurr1 can increase expression of claustrum marker genes weakens the conclusion that Nurr1 is a "master switch for the gene transcription profile of claustrum". In the least, the "master" language should be heavily tempered, especially considering that other transcription factors were not knocked out to compare with the effects of the Nurr1 deletion.
7. The claustrum is known to be necessary for performance of tasks requiring particular

cognitive demand and/or attention (see work from Citri, Mathur, and Seminowicz labs). As such, it is no surprise that no effect of Nurr1 deletion was observed on behavior (i.e. because the claustrum wasn't being behaviorally assayed in the first place). Using behaviors demonstrated in the literature to assay claustrum function would more meaningfully address this question.

8. Figure 7. The time scale shown in figure 7i,o only captures the first 15 seconds after injection – arguably insufficient to capture the effects of the drug on functional connectivity and inexplicably only a fraction of the time-span used in figures 7f,g,i,l,m. Further it is not clear over what time span the AUC results are from in figures 7f,g,i,l,m,o. Finally, while figures 7h and 7n show the DOI and U69 increases functional connectivity on a minute time scale, which is observed in the control mice but not in the D1R-Nurr1cKO mice, this effect may be entirely due to the difference in statistical power resulting from a sample size of control mice that is almost 3-fold that of D1R-Nurr1cKO mice (15 vs 6). Repeating this experiment in AAV-Nurr1cKO mice using an equivalent number of control and cKO mice and using comparisons across equal time spans is critical to determine if claustrum Nurr1 is necessary for hallucinogen-induced increase in functional connectivity.

9. The discussion section should include a discussion of the limitations of this study including, but not limited to, whether cell packing density differences in claustrum vs. surrounding regions may influence ISH optical density measures.

Minor

1. Some language about the function of the claustrum in the intro is a bit outdated. The Crick model has not received any supporting evidence beyond an N=1 clinical study that was not replicated.

2. Saying that claustrum mainly targets PV and NPY interneurons in cortex is perhaps not completely accurate (see PMIDs: 29623428 and 36368317). Perhaps just saying that claustrum can, in some cortical areas, strongly innervate these neuron types would be best.

3. One cannot conclude that the Nurr1 KO that leads to decreased 5HT2R receptor expression gives rise to a decrease in fos expression following DOI challenge. Please modulate language. Several circuit explanations may give rise to this effect that would have nothing to do with 5HT2R expression in CLA.

4. Lines 195 to 204 don't add much and could be removed for brevity sake.

5. The study further shows that Nurr1 is required for expression of 5-HT2AR, 5-HT2CR, and KOR – receptors targeted by psychedelics and salvia and known to be involved in producing the subjective experience induced by these psychoactive drugs. This finding is confirmed functionally through DOI-induced expression of immediate early genes and KOR ligand-induced depression of claustrum fEPSPs. However, the claim that claustrum expression of these receptors regulates the hallucinogenic-like state is a bit aggressive, especially in the absence of HTR effects.

6. Line 122-123 states that Nurr1 expression in D1R-Nurr1cKO was maintained in SNc/VTA and hippocampus. However, in the referenced figure 4f-4j Nurr1 expression is significantly reduced in both SNc and subiculum. Change language to accurately reflect the data.

7. Line 162-163 states that Nurr1 plays a pivotal role in maintaining functional identity of claustrum neurons. However, there is no functional data presented in the referenced figure 5.

8. It would be nice to see vehicle effects in figure 7.

9. A detailed description of how claustrum was identified in control and Nurr1cKO mice is necessary to properly evaluate the validity of the ISH experiments (related to major point 1).

10. What genetic mouse-line was used as control for the D1R-Nurr1cKO mice? D1R-Cre

mice or flx-Nurr1 mice?

Reviewer #2 (Remarks to the Author):

Summary:

In this manuscript, Mantas et al. investigate the role of Nurr1 in claustrum neuron identity and how it mediates hallucinogenic experience. Specifically, they demonstrate ablation of Nurr1 results in aberrant gene expression profiles in the claustrum, but does not affect claustrrocortical projections. Further, they demonstrate ablation of Nurr1 eliminates claustrum-driven hallucinogenic experiences.

Overall, the experiments are well-executed and authors present solid and well-validated findings in many cases, and I commend the authors for this extensive body of work. However, the manuscript does not really read like a cohesive paper; rather, it feels like a series of experiments that are loosely related and don't have logical cohesion. I think there is some important information here for the claustrum community, but in this reviewer's opinion the manuscript needs to be significantly revised to have a clear narrative prior to potentially being accepted for publication.

Major Comments:

1. Logical flow: The major ideas throughout the paper are not well-connected, and as such the overall aim and big picture finding is not clearly communicated. One example of this is the introduction, where the topic jumps from CLA connectivity, to the role of CLA in hallucinogenic experience, to the role of Nurr1 in dopaminergic neuron development, to Nurr1 in claustrum development, to other topics. These concepts should be better connected, as the reader is not well-oriented to understand the key background nor core goal of manuscript. Even the abstract itself is jumpy and difficult to understand.
2. Related to (2), many of the results presented seem off-target or lack relevance to the key parts of the manuscript. The findings that Nurr1 regulates claustrum transcriptional identity are rock solid and logically and carefully validated, and the authors did a wonderful job here. However, this same care is not done for many other parts of the manuscript, wherein results seemingly appear out of nowhere and do not relate to the main theme (examples include but not limited to: cross-species work in Figure 1, retention and analysis of non-claustrum spots in spatial transcriptomics datasets of Figures 2 and 3, behavioral characterization of Figure 4). These tangential datasets make it hard to follow the core message of the paper, and make the paper feel like cataloging rather than being directed.
3. The authors need to carefully demonstrate the D1R-Nurr1 strategy is selective for claustrum neurons. Certainly this manipulation does capture claustrum neurons, but D1R is highly expressed in other adjacent cell types and thus this manipulation may lead to off-target effects. The authors need to show the extent of specificity to claustrum neurons, otherwise it is difficult to interpret the results with any sort of claustrum-specific role.
4. At several points in the manuscript, the role of Nurr1 in the development of CLA is highlighted, however the results do not corroborate this perspective. A key component of developmental papers is that they investigate their hypothesis at critical developmental stages and compare differences across these timepoints, as seen in (Fang et al., Front

Neuroanat 2021) where they examine the developmental progression of claustrum Nurr1 neurons. I would advise the authors to take caution when using the word development, as it doesn't seem like any results in this paper have to do with development per se. However, if the results do relate to development, they need to be reframed so the reader can better understand this.

a. In this study it is stated that mice 2-6 months of age were used and by ~2 months of age, the mouse brain is considered to be mature.

b. In the abstract, it is mentioned that Nurr1's role in the developing and adult CLA is not understood, however the results do not investigate the role of Nurr1 in the developing CLA specifically, but the mature CLA.

c. In the results section, it is stated that the authors were investigating how Nurr1 influences development of CLA neurons by using D1R-cre mice to knock out Nurr1 early in CLA neurons. It would be informative to indicate at what timepoint the knockout has occurred.

d. In the discussion, it is stated that "CLA Nurr1 deletion in both early and adult stage life stages, produces a robust reduction of CLA enriched genes." In previous sections of the paper and/or in the methods early and adult life stages is not defined. Is early life considered embryonic, neonatal, adolescent stages?

Minor Comments:

1. For the open field test, only total distance travelled was measured. However, other measurements may also be informative of the presence of hyperactive behaviours, including time spent in the inner and outer zones of OF and/or speed.

2. Figure 4j caption states the n = 3-5. An exact n value provided would be informative.

3. Stylistic comments: It would be helpful for the reader if each figure was annotated as Fig 1, 2, 3 etc.

4. Figure 3g-v is very condensed and hard to read. Better alignment of the ISH autoradiographs and corresponding graphs would make the panel easier to read.

5. In the methods it states Sprague-Dawley rats were used, however there is no reference to rats in each of the experiments. Clarification regarding which experiments used mice versus rats would be helpful.

6. In the results section where behavioural results are discussed, a concluding sentence is missing. Additionally, when discussing the results of the OFT it would be informative to include the purpose of AMPH and PCP in the experiment. (i.e. AMPH and/or PCP were administered to increase dopaminergic neurotransmission during the OFT which.....)

7. For the spatial transcriptomics analysis, it would be better to exclude all non-claustrum spots and perform analysis solely on the claustrum. Other spots will exert control over the global and local distancing within the UMAP embedding, and this distorts the relationship between claustrum neurons, the focal point of this paper.

8. Figure 2h seems to use two different scale bars.

9. There are many other marker genes for claustrum neurons, and neuron subtypes (Erwin et al., eLife 2021). Comparisons to these cell types and transcriptomes, rather than just the genes obtained from the Allen ISH website, would likely be informative.

Reviewer #3 (Remarks to the Author):

The manuscript by Mantas et al. reports investigations into the developmental regulation of the claustrum (CLA) and the function of the CLA in several behaviors in the mouse. A wide

variety of techniques ranging from ISH to electrophysiology to single cell transcriptomics to behavioral pharmacology are performed in the analysis. The manuscript is extremely well written, describing well planned experiments that have resulted in high quality data supporting their general conclusions. Their overall conclusions support the notion that Nurr1 is critical for the development and function of the claustrum in mice, and that the claustrum is involved in mediating functional connectivity of brain regions similar to that of humans in response to hallucinogens of the psychedelic and KOR classes. Their use of fUS to investigate mouse brain functional connectivity is a strength, demonstrating proof of principle for future use of this technique to investigate connectivity in response to genetic/pharmacological manipulations in rodent brain. Overall the findings and conclusions of the work are of high impact towards our understanding of fundamental brain mechanisms underlying behaviors, and mechanism of action of hallucinogens. I only have a few minor concerns:

DOI should be defined as: 2,5-dimethoxy-4-iodoamphetamine.

Although the ISH data are convincing that CLA enriched expression is reduced, no antibody confirmation experiments were performed (see Fig 5).

As I am sure the authors are aware, gene expression does not always correlate to protein expression and a disclaimer should be included in the results section of this limitation.

Is there any speculation as to the mechanism of how Nurr1 activity is involved in specification of the CLA?

HTR is largely a reflex of activation of cortical 5-HT_{2A} receptors, and this was unchanged by CLA ablation. A more interesting behavioral experiment to perform would be 2-lever drug discrimination as this probes the introspective cue of psychoactive drugs. Would ablating CLA function in adults leave HTR intact, but remove the ability to discriminate the subjective effects of hallucinogens? Being able to separate these effects in an experimental system would be a major development in the study of psychedelics.

Although not a concern, this is more of a comment. It would be fascinating to use intersectional genetics to express DREADD receptors, especially the KORD, in the CLA for conditional manipulation of activity in adults in behavioral experiments.

Reviewer #4 (Remarks to the Author):

- In this study, Mantas et al., investigated the role of Nurr1 as a central regulator of hallucinogenic effect. The study is well conducted, the manuscript well written. The part of the study using ultrasound imaging as a readout for neural correlate is well thought and statistical analysis seem appropriate. There are, however, several points (listed below) to consider in order to strengthen and clarify the study.

- Authors should state more clearly what is the rationale for the use of ultrasound imaging in their study. This technique of ultrafast Doppler can be used to measure different modalities. They introduce the concept of "neuronal correlate". It is true that due to neurovascular coupling, when animals are exposed to a task, changes in CBV are due to increased neuronal activation. However, this is not always the case during a task, as recently reviewed by Patrick Drew (<https://doi.org/10.1016/j.tins.2022.08.004>). In addition, in pharmacological

studies, things are far more complicated. Pharmacologically active drugs may have a direct unspecific vascular effect, which has nothing to do with neural correlate; or can be in addition to it.

In addition, here the study does not use functional ultrasound, as they did not do any activations per se. They rather performed pharmacological fUS, which is different. This only appears in the discussion section.

Authors should mention these concepts, correct and clarify their text accordingly.

In addition, authors should quantify the unspecific vascular effect induced by the drugs tested, using variations of CBV in the medial vein and in one artery. They are both clearly visible on panel C of figure 7.

- One major issue of this study is the low Doppler signal in the anterior part of the brain in the adult mice, especially in the lateral part. It is very difficult and scientifically inaccurate to study changes if the signal is weak/inexistent. This is particularly problematic for the SM that seems to have no signal at all. There is no problem in medial structures, however.

- What is the rationale for the choice of regions of interest to study? In particular, why use overlapping areas? Only at the end of the discussion, the reader understands your motivation: to compare in your animal model what other teams described in human (decreased CBV in some of these ROI and increased functional connectivity between PFC and SM). This should be stated earlier in the manuscript.

Minor points

- Some of your figures or figure legends are not accurate enough. For instance:
- It is unclear from the figure legend if the results presented in panel E are an example or an average over X animals.
- What is the X axis of the panel E? Is it time? This is not clear.
- In panels f and g: Why is there a pink rectangle with the mention 'PFC'? The reviewer understood that these were the changes of CBV in the ACA(f) and MO (g). But not in the PFC. This is unclear.

Response to Referees

Reviewer #1 (Remarks to the Author):

Mantas et al. present a rather interesting study on the role of the transcription factor Nurr1 in regulating claustrum development and the ability for the claustrum to contribute to hallucinogenic states. They extensively show that conditional knock-out of the transcription factor Nurr1 alters gene expression in the claustrum including reduced expression of claustrum enriched genes. They further show that conditional Nurr1 knock-out reduces expression of receptors known to mediate hallucinogen-induced subjective experiences 5-HT2AR, 5-HT2CR, and KOR and confirm this finding functionally. Finally, they show that agonists of 5-HT2AR and KOR increases functional connectivity between prefrontal cortex and sensorimotor cortices in a Nurr1-dependent manner. Several points require addressing to improve the manuscript.

Major

1. Nurr1 expression is apparent in the claustrum, but also in the ventrally-lying endopiriform nucleus as well as what looks like layer 6b neurons lining the lateral and dorsal aspects of the external capsule/corpus collosum. Receptor expression changes are not assessed in these areas to help address whether the fc findings are secondary to CLA receptor expression changes, or changes elsewhere. In addition, what independent verification exists to say that clusters 8 and 11 are the same anatomical region? 8 genetically looks like claustrum, but it could be that 11 is endopiriform nucleus. This needs to be addressed.

Response:

Thank you for your comment. Based on our data on Nurr1 mRNA distribution, it is evident that the expression extends to cortical layer 6b and the ventrally located dorsal endopiriform nucleus (DEn). Our AAV-Cre approach leads to Nurr1 deletion in both layer 6b and DEn. Our single-cell transcriptomics (ST) dataset reveals that layer 6b belongs to a different cluster (cluster 12) than the claustrum (CLA) (cluster 8).

To further analyze this, we performed a deconvolution analysis using the Telencephalon Excitatory Projecting Neurons spatial dataset from <https://doi.org/10.1016/j.cell.2018.06.021> (Figure S5). In this dataset, the CLA belongs to TEGLU20 and TEGLU12, while layer 6b is mostly represented by TEGLU2. We have added UMAPs in Figure 2, showing the absence of

TEGLU20 and TEGLU12 in AAV-Nurr1cKO. Additionally, we have included supplementary figures showing the distribution of TEGLU2 in Ctrl and AAV-Nurr1cKO.

Since there is high expression of D1R in layer 6b, our D1R-Nurr1cKO mice display a loss of Nurr1 in layer 6b as well (Figure 4). Considering that 5HT2AR is also enriched in layer 6b (but not kOR), we assessed the 5HT2AR expression in layer 6b of D1R-Nurr1cKO mice. Alongside 5HT2AR, we examined the expression of a layer 6b/CLA marker, Tmem163. Although D1R-Nurr1cKO mice display a notable decrease in 5HT2AR in the CLA, this effect is not observed in layer 6b. This result is consistent with the fact that D1R-Nurr1cKO mice show a dramatic decrease in Tmem163 in the CLA but no change in Tmem163 expression in layer 6b.

As for the DEn, literature (<https://doi.org/10.1002/cne.24537>) indicates that it is very hard to transcriptionally distinguish the CLA from the DEn. It is stated that the CLA and DEn are parts of the same complex. However, there are genetic, neurochemical, and architectural differences between the CLA and DEn, suggesting they may be functionally separate nuclei within the same complex. Deconvolution analysis from <https://doi.org/10.1016/j.cell.2018.06.021> (Figure S5) failed to separate the CLA from the DEn in our dataset. Notably, cluster 11, which defines the claustral region in AAV-Nurr1cKO, did not belong to any of the TEGLU clusters.

Nevertheless, we decided to change the term "claustrum" to "claustral complex," which includes both the claustrum and DEn. This term is used for these structures in the following publications: <https://doi.org/10.1002/cne.24537>, <https://doi.org/10.1002/cne.25539>.

2. The authors use an indirect method of examining whether claustracortical projections remain intact in the absence of Nurr1. Would suggest injecting rAAV-cre-gfp into a medial PFC of Nurr1-flx and wt mice and compare the number of retrogradely labeled neurons.

Response:

Excellent recommendation! We injected rgAAV-Cre into the right medial PFC and AAV-DIO-GFP into the right CLA of Nurr1-flx and wild-type mice. We did not observe any differences between the two genotypes. We have added Supplementary Fig. 6 to show the results of this experiment.

3. HTR is the gold standard for behavioral assessment of psychedelic (2a agonist) drug challenge in rodents. The lack of HTR changes following Nurr1 KO would strongly suggest that CLA 2a receptor has nothing to do with HTR, as the authors state, but also suggests that the CLA then has nothing to do with psychedelic states. This stands in conflict with the primary

conclusions of this paper. Claiming that HTR is not a good readout because KOR agonists don't elicit them is not cogent, as KOR is a completely different receptor than 5HT_{2a} (unrelatedly, agonists to 2a are considered psychedelics and agonists of KOR are considered hallucinogens). The subjective responses to these drugs are vastly different, along vastly different timescales in humans.

Response:

Thank you for your comment. While we acknowledge that the head-twitch response (HTR) is commonly used as a benchmark for characterizing psychedelic drugs, it remains uncertain whether the HTR accurately reflects the hallucinogenic effects of these substances. This uncertainty is underscored by the observation that certain non-hallucinogenic drugs, such as 5-HTP, the 5-HT releasing agent fenfluramine, and the cannabinoid receptor antagonist SR141716A, induce HTR in rodents despite lacking hallucinogenic activity in humans (<https://doi.org/10.1002/dta.1333>, <https://doi.org/10.1007/BF00401509>). Nevertheless, we recognize that the HTR may capture certain aspects of the hallucinogenic experience that are specific to 5HT_{2A}R agonists and, according to our data, may not be influenced by claustral 5HT_{2A}R. We have included a relevant discussion on this topic.

4. AAV-Nurr1cKO was not used to bolster the D1R-Nurr1cKO fUS findings. This is a major weakness as indicated by the lack of HTR in the D1R-Nurr1cKO mice, which indicates that the “hallucinogenic” effects of DOI may lie outside the claustrum.

Response:

Thank you for your insightful comment. Unfortunately, we were unable to reuse the equipment utilized for fUS imaging due to administrative issues. As a result, we opted for an alternative approach to strengthen our findings, which we could perform at Karolinska Institute. We decided to assess functional connectivity through Neuropixels recordings. Specifically, we recorded simultaneously with two Neuropixels probes (one in the PFC and one in the somatosensory cortex) in Ctrl, D1R-Nurr1cKO, and AAV-Nurr1cKO mice after administering saline or 1 mg/kg of DOI.

When comparing fUS and Neuropixel probe electrophysiological recordings, it is important to note that hemodynamic responses differ substantially from electrophysiological changes. Since we propose a role for the CLA in cortical synchrony following hallucinogen administration, our primary goal was to identify correlation changes in Ctrl mice that differ from those in D1R-

Nurr1cKO and AAV-Nurr1cKO mice. Considering that EEG studies on psychedelics report desynchronization (<https://doi.org/10.1523/JNEUROSCI.2063-13.2013>, <https://doi.org/10.1038/s41598-019-51974-4>, <https://doi.org/10.1038/s41398-021-01603-4>) rather than an increase in functional connectivity, we hypothesized that Neuropixels recordings would show a loss of coherence between the PFC and SM. Interestingly, we found that DOI reduced the correlation between the MOs (PFC) units and the SSp (SM) units (Fig. 7q). This effect was significantly different from the effects observed in DIR-Nurr1cKO and AAV-Nurr1cKO mice (Fig. 7q). Moreover, there was no significant difference between DIR-Nurr1cKO and AAV-Nurr1cKO mice (Fig. 7q), underscoring the role of the CLA in global cortical connectivity changes following DOI administration.

We also recognize that the HTR is a well-established hallmark of hallucinatory behavior in mice, as noted in the literature. Both DIR-Nurr1cKO and AAV-Nurr1cKO mice exhibit this behavior. Therefore, we have toned down the language and mentioned in the discussion that the CLA might play a role rather than being the principal brain area mediating the hallucinatory effects.

5. It is important to experimentally consider what would happen to CLA immediate early gene expression and fUS effects if a non-hallucinogenic drug were administered (eg meth, cocaine, ethanol or heroin). If nothing was found with these drug classes, this would strengthen the claim that the CLA is involved in hallucination, rather than non-specific drug responses writ large.

Response:

We greatly appreciate your comment. It is important to note that dopamine-releasing agents such as methamphetamine and morphine have been documented to cause hallucinations in humans (<https://doi.org/10.1213/ANE.0000000000001417>, <https://doi.org/10.3389/fpsyt.2021.694863>). In line with this, D2 receptor antagonists have been reported to block the occurrence of hallucinations. Additionally, there is literature (<https://doi.org/10.1016/j.cub.2020.06.064>) indicating that chronic administration of high doses of cocaine (20 mg/kg) stimulates D1Rs in the claustrum and increases immediate early gene expression in this region. High doses of cocaine have also been reported to exhibit hallucinogenic effects in humans (<https://doi.org/10.3389/fpsyt.2021.694863>).

In response to your comment, we decided to examine Arc expression in the CLA of Ctrl and DIR-Nurr1cKO mice after administering lower doses of cocaine. We chose this dose because it has been reported to increase immediate early gene expression in other brain areas, such as

the dorsomedial striatum (<https://doi.org/10.1038/sj.npp.1300832>). Since Nurr1 deletion produced a robust decrease in the expression of numerous CLA genes, we first examined the expression of D1R in D1R-Nurr1cKO mice. Interestingly, the CLA D1R mRNA signal was not affected by Nurr1 deletion.

Despite this, one week of daily cocaine administration did not produce any significant change in Arc expression in the CLA of Ctrl and D1R-Nurr1cKO mice. However, the same brains displayed a significant increase in Arc expression in the dorsomedial striatum. Additionally, these mice showed increased locomotion in the open field compared to their saline-treated littermates in both genotypes, indicating that the loss of Nurr1 in the CLA does not affect the locomotor response to subchronic medium doses of cocaine. These data are shown in Supplementary Fig. 10.

In response to your concerns about the fUS effects, we have added data from previous experiments using the 5HT2R antagonist ketanserin and the κ OR antagonist nor-binaltorphimine. Pretreatment with these drugs reduced the fUS-fc effects observed after DOI and U69 administration, respectively. These findings are now detailed in Supplementary Fig. 11. We hope these additional experiments address your concerns.

6. In figure 3 and 5, transcriptomics and in-situ hybridization are used to convincingly show that Nurr1 knock-out results in reduced expression of several genes enriched in the claustrum. However, the lack of a sufficiency experiment showing that Nurr1 can increase expression of claustrum marker genes weakens the conclusion that Nurr1 is a “master switch for the gene transcription profile of claustrum”. In the least, the “master” language should be heavily tempered, especially considering that other transcription factors were not knocked out to compare with the effects of the Nurr1 deletion.

Response:

We appreciate your comment and agree with your concerns. We removed the term “master switch” from our manuscript and we toned down the language that we use to describe the effects.

7. The claustrum is known to be necessary for performance of tasks requiring particular cognitive demand and/or attention (see work from Citri, Mathur, and Seminowicz labs). As such, it is no surprise that no effect of Nurr1 deletion was observed on behavior (i.e. because the claustrum wasn't being behaviourally assayed in the first place). Using behaviors

demonstrated in the literature to assay claustrum function would more meaningfully address this question.

Response:

Thank you for your comment. Recently, there have been several publications studying the role of CLA. Particularly, there is evidence that CLA lesions produce a depressive-like phenotype (increased immobility time in FST and reduced sucrose consumption in SPT: <https://doi.org/10.1038/s41467-023-43636-x>) and impairs pain-avoidance learning (impaired pain learning to 1% acetic acid in a conditioned place aversion paradigm: <https://doi.org/10.1016/j.cub.2024.03.044>). In this study, we assessed the performance of D1R-Nurr1cKO mice in FST, SPT and PAT (pain avoidance learning paradigm), which was similar to their Ctrl littermates. These surprising results bolster the notion that Nurr1 deletion does not impair substantially the baseline activity of CLA. We moved these data to the main Fig. 4.

8. Figure 7. The time scale shown in figure 7i,o only captures the first 15 seconds after injection – arguably insufficient to capture the effects of the drug on functional connectivity and inexplicably only a fraction of the time-span used in figures 7f,g,l,m. Further it is not clear over what time span the AUC results are from in figures 7f,g,i,l,m,o. Finally, while figures 7h and 7n show the DOI and U69 increases functional connectivity on a minute time scale, which is observed in the control mice but not in the D1R-Nurr1cKO mice, this effect may be entirely due to the difference in statistical power resulting from a sample size of control mice that is almost 3-fold that of D1R-Nurr1cKO mice (15 vs 6). Repeating this experiment in AAV-Nurr1cKO mice using an equivalent number of control and cKO mice and using comparisons across equal time spans is critical to determine if claustrum Nurr1 is necessary for hallucinogen-induced increase in functional connectivity.

Response:

Thank you for your valuable observation. We apologize for the incorrect labeling of the x-axis, which has now been corrected (Fig. 7g, l). Additionally, we have included the data from the vehicle portion of the recording (Fig. 7g, l). We agree that the sample size differences may impact the correlation matrix results. Consequently, we have removed the correlograms and retained the direct comparison of effects in Ctrl and D1R-Nurr1cKO mice (Fig. 7g, l).

The specificity of these effects is underscored by the fact that the same number of animals did not reach significance in CBV changes, with both groups showing a profound signal reduction.

As mentioned earlier, due to administrative constraints, we were unable to perform additional fUS experiments. Therefore, we proceeded with neuropixel recordings using a similar sample size (animal n and unit n are detailed in Fig. 7 legend), including AAV-Nurr1cKO mice.

Based on previous EEG studies in humans, we anticipated a reduced rather than increased correlation between PFC and SM units. Our observations indicate that DOI decreased the correlation between MOs (PFC) units and SSp (SM) units (Fig. 7q). This effect was significantly different from the effects observed in D1R-Nurr1cKO and AAV-Nurr1cKO mice (Fig. 7q). Furthermore, there was no significant difference between D1R-Nurr1cKO and AAV-Nurr1cKO mice (Fig. 7q), highlighting the role of the CLA in global cortical connectivity changes following DOI administration.

9. The discussion section should include a discussion of the limitations of this study including, but not limited to, whether cell packing density differences in claustrum vs. surrounding regions may influence ISH optical density measures.

Response:

Thank you for your suggestion. We understand the concern that the packing density in the CLA is higher than in surrounding structures, and that any change in cell packing due to Nurr1 deletion might affect the ISH optical density measurements. To address this, we included ISH data using a probe for the non-coding mRNA of Nurr1 (Supplementary Fig. 7). Our results show that the signal for the non-coding Nurr1 mRNA remains intact, indicating no cell packing deficit after Nurr1 deletion. Additionally, we have added a discussion section to acknowledge some limitations of the study.

Minor

1. Some language about the function of the claustrum in the intro is a bit outdated. The Crick model has not received any supporting evidence beyond an N=1 clinical study that was not replicated.

Response:

We agree with your point. The review in question is a hallmark piece that has significantly attracted the scientific community's attention to the study of the CLA's function. Therefore, we have added an introduction about the CLA's function, incorporating up-to-date citations.

2. Saying that claustrum mainly targets PV and NPY interneurons in cortex is perhaps not completely accurate (see PMIDs: 29623428 and 36368317). Perhaps just saying that claustrum can, in some cortical areas, strongly innervate these neuron types would be best.

Response:

Thank you for this comment. We changed the text accordingly.

3. One cannot conclude that the Nurr1 KO that leads to decreased 5HT2R receptor expression gives rise to a decrease in fos expression following DOI challenge. Please modulate language. Several circuit explanations may give rise to this effect that would have nothing to do with 5HTR2 expression in CLA.

Response:

Thank you for your insightful comment. We have modified the text and also included immunofluorescent images showing the internalization of 5HT2AR after DOI administration in the CLA of Ctrl and D1R-Nurr1cKO mice (Fig. 6e, f). This provides a more direct demonstration of the effect of DOI on 5HT2AR than the upregulation of Fos or Arc.

4. Lines 195 to 204 don't add much and could be removed for brevity sake.

Response:

Thank you. We modified the text according to your suggestions.

5. The study further shows that Nurr1 is required for expression of 5-HT2AR, 5-HT2CR, and KOR – receptors targeted by psychedelics and salvia and known to be involved in producing the subjective experience induced by these psychoactive drugs. This finding is confirmed functionally through DOI-induced expression of immediate early genes and KOR ligand-induced depression of claustrum fEPSPs. However, the claim that claustrum expression of these receptors regulates the hallucinogenic-like state is a bit aggressive, especially in the absence of HTR effects.

Response:

Thank you. We modified the text according to your suggestions.

6. Line 122-123 states that Nurr1 expression in D1R-Nurr1cKO was maintained in SNc/VTA and hippocampus. However, in the referenced figure 4f-4j Nurr1 expression is significantly reduced in both SNc and subiculum. Change language to accurately reflect the data.

Response:

Insightful comment. We modified the text according to your suggestions.

7. Line 162-163 states that Nurr1 plays a pivotal role in maintaining functional identity of claustrum neurons. However, there is no functional data presented in the referenced figure 5.

Response:

Great observation. We modified the text according to your suggestions.

8. It would be nice to see vehicle effects in figure 7.

Response:

Thank you for your suggestion. We have incorporated the recordings, including the vehicle part, into Fig. 7d-f and Fig. 7i-l.

9. A detailed description of how claustrum was identified in control and Nurr1cKO mice is necessary to properly evaluate the validity of the ISH experiments (related to major point 1).

Response:

Thank you for your suggestion. To identify the CLA, we followed a similar approach to the study found here: <https://doi.org/10.1101/2022.06.02.494429>. In Fig. 1a, we have added yellow rectangles centered on the CLA to illustrate how we made the measurements. Additionally, we have included a detailed description of our approach in the 'Radioactive In Situ Hybridization' section of the Methods.

10. What genetic mouse-line was used as control for the D1R-Nurr1cKO mice? D1R-Cre mice or flx-Nurr1 mice?

Response:

The mouse line that was used as Ctrl was mice that were homozygous for Nurr1-flx. In Fig. 4 we have illustration showing the genetic construct of Ctrl mice. For fUS experiments we used both Nurr1-flx and WT mice (neither Nurr1-flx or D1R-Cre). We added relevant text in the 'Animals' section of Methods.

Reviewer #2 (Remarks to the Author):

Summary:

In this manuscript, Mantas et al. investigate the role of Nurr1 in claustrum neuron identity and how it mediates hallucinogenic experience. Specifically, they demonstrate ablation of Nurr1 results in aberrant gene expression profiles in the claustrum but does not affect claustrrocortical projections. Further, they demonstrate ablation of Nurr1 eliminates claustrum-driven hallucinogenic experiences.

Overall, the experiments are well-executed and authors present solid and well-validated findings in many cases, and I commend the authors for this extensive body of work. However, the manuscript does not really read like a cohesive paper; rather, it feels like a series of experiments that are loosely related and don't have logical cohesion. I think there is some important information here for the claustrum community, but in this reviewer's opinion the manuscript needs to be significantly revised to have a clear narrative prior to potentially being accepted for publication.

Major Comments:

1. Logical flow: The major ideas throughout the paper are not well-connected, and as such the overall aim and big picture finding is not clearly communicated. One example of this is the introduction, where the topic jumps from CLA connectivity to the role of CLA in hallucinogenic experience, to the role of Nurr1 in dopaminergic neuron development, to Nurr1 in claustrum development, to other topics. These concepts should be better connected, as the reader is not well-oriented to understand the key background nor core goal of manuscript. Even the abstract itself is jumpy and difficult to understand.

Response:

Thank you for your comment. We have modified the manuscript according to your suggestions and hope that the text now reads well.

2. Related to (2), many of the results presented seem off-target or lack relevance to the key parts of the manuscript. The findings that Nurr1 regulates claustrum transcriptional identity are rock solid and logically and carefully validated, and the authors did a wonderful job here. However, this same care is not done for many other parts of the manuscript, wherein results seemingly appear out of nowhere and do not relate to the main theme (examples include but not limited to: cross-species work in Figure 1, retention and analysis of non-claustrum spots in spatial

transcriptomics datasets of Figures 2 and 3, behavioral characterization of Figure 4). These tangential datasets make it hard to follow the core message of the paper, and make the paper feel like cataloging rather than being directed.

Response:

We completely understand your concerns. The cross-species work in Fig. 1 supports the translational nature of our study. With Figure 1, we aim to demonstrate that Nurr1 shows similar enrichment in the CLA across species, providing evidence that a possible Nurr1 deletion in human brains would lead to similar results as in mice. This is further supported by the fact that we used the same Nurr1 probe to detect Nurr1 mRNA in all species, underscoring the high interspecies homology of the transcription factor. We have modified the text to clarify the core message of this work.

Regarding the ST dataset, excluding non-claustrum spots would be relevant only if we were focusing on identifying CLA subclusters. Conducting clustering analysis without excluding any spots allowed us to identify CLA as a transcriptionally distinct cluster, which was not present after the deletion of Nurr1. Isolating only claustrum spots would introduce bias to our results. Moreover, including all spots allows us to show that the separate cluster that occurs in AAV-Nurr1cKO is unique and does not share transcriptional features with other defined clusters in our sections.

3. The authors need to carefully demonstrate the D1R-Nurr1 strategy is selective for claustrum neurons. Certainly, this manipulation does capture claustrum neurons, but D1R is highly expressed in other adjacent cell types and thus this manipulation may lead to off-target effects. The authors need to show the extent of specificity to claustrum neurons, otherwise it is difficult to interpret the results with any sort of claustrum-specific role.

Response:

Excellent comment. We are aware that D1R is expressed in adjacent cell types such as CPu and L6b. We did not detect any Nurr1 expression in CP. However, there is notable co-expression of Nurr1 with D1R in L6b. We have added a quantification of Nurr1 expression in L6b of D1R-Nurr1cKO mice, which showed a substantial reduction of Nurr1 in L6b of this mouse line (Fig 4).

We also examined whether the effects of Nurr1 loss in L6b are similar to those in CLA. In situ hybridization (ISH) experiments with an anti-sense Tmem163 probe, a specific marker for L6b-

CLA, showed a robust reduction of *Tmem163* in CLA but not in L6b of the D1R-Nurr1cKO mice (Supplementary Fig. 8). Since our main outcomes stem from the loss of 5HT_{2A}R/5HT_{2C}R and κOR in CLA, we examined the expression of these receptors in D1R-Nurr1cKO mice. From our ISH images, KOR and 5HT_{2C}R do not show any L6b expression (Fig. 5). However, 5HT_{2A}R is enriched in L6b, and we quantified its mRNA signal in D1R-Nurr1cKO mice (Supplementary Fig. 8). Similar to *Tmem163*, 5HT_{2A}R was not reduced in L6b of D1R-Nurr1cKO mice, supporting our claim that CLA may affect hallucinogen-induced connectivity effects (Supplementary Fig. 8).

Moreover, we complemented our connectivity studies with Neuropixel recordings, including both D1R-Nurr1cKO and AAV-Nurr1cKO mice (Fig. 7). Considering that EEG studies on psychedelics report desynchronization rather than an increase in functional connectivity (<https://doi.org/10.1523/JNEUROSCI.2063-13.2013>, <https://doi.org/10.1038/s41598-019-51974-4>, <https://doi.org/10.1038/s41398-021-01603-4>), we hypothesized that Neuropixel recordings would show a loss of coherence between the PFC and SM. Interestingly, we found that DOI reduced the correlation between the MOs (PFC) units and the SSp (SM) units (Fig. 7q). This effect was significantly different from the effects observed in D1R-Nurr1cKO and AAV-Nurr1cKO mice. Furthermore, there was no significant difference between D1R-Nurr1cKO and AAV-Nurr1cKO mice, indicating that our results in D1R-Nurr1cKO mice may not stem from off-target deletion of *Nurr1* in adjacent structures.

4. At several points in the manuscript, the role of *Nurr1* in the development of CLA is highlighted, however the results do not corroborate this perspective. A key component of developmental papers is that they investigate their hypothesis at critical developmental stages and compare differences across these timepoints, as seen in (Fang et al., Front Neuroanat 2021) where they examine the developmental progression of claustrum *Nurr1* neurons. I would advise the authors to take caution when using the word development, as it doesn't seem like any results in this paper have to do with development per se. However, if the results do relate to development, they need to be reframed so the reader can better understand this.

a. In this study it is stated that mice 2-6 months of age were used and by ~2 months of age, the mouse brain is considered to be mature.

b. In the abstract, it is mentioned that *Nurr1*'s role in the developing and adult CLA is not understood, however the results do not investigate the role of *Nurr1* in the developing CLA specifically, but the mature CLA.

c. In the results section, it is stated that the authors were investigating how Nurr1 influences development of CLA neurons by using D1R-cre mice to knock out Nurr1 early in CLA neurons. It would be informative to indicate at what timepoint the knockout has occurred.

d. In the discussion, it is stated that “CLA Nurr1 deletion in both early and adult stage life stages, produces a robust reduction of CLA enriched genes.” In previous sections of the paper and/or in the methods early and adult life stages is not defined. Is early life considered embryonic, neonatal, adolescent stages?

Response:

Thank you for your comment. We have removed the word 'development' wherever we refer to the CLA. Additionally, we have included new data showing the expression of Nurr1 in the CLA of postnatal day 1 D1R-Nurr1cKO pups, where Nurr1 was already ablated at that stage (Supplementary Fig. 7). However, we have chosen not to mention development in the manuscript.

Minor Comments:

1. For the open field test, only total distance travelled was measured. However, other measurements may also be informative of the presence of hyperactive behaviours, including time spent in the inner and outer zones of OF and/or speed.

Response:

Thank you for this comment. We have addressed this by plotting the speed instead of the total distance travelled, as shown in Supplementary Figure 10.

2. Figure 4j caption states the n = 3-5. An exact n value provided would be informative.

Response:

Thank you for your suggestion. We have added the exact n value and changed the bar plots to box plots to make the information more informative.

3. Stylistic comments: It would be helpful for the reader if each figure was annotated as Fig 1, 2, 3 etc.

Response:

Thank you for this suggestion. We have labeled the figures in the top left corner as Fig. 1, Fig. 2, and so on.

4. Figure 3g-v is very condensed and hard to read. Better alignment of the ISH autoradiographs and corresponding graphs would make the panel easier to read.

Response:

Thank you for your great suggestion. We have aligned the ISH autoradiographs and the corresponding graphs. We hope this will work better for you.

5. In the methods it states Sprague-Dawley rats were used, however there is no reference to rats in each of the experiments. Clarification regarding which experiments used mice versus rats would be helpful.

Response:

Great observation. The rats were only used in the radioactive ISH in Fig1. We have pinpointed this in the 'Animals' section of Methods.

6. In the results section where behavioural results are discussed, a concluding sentence is missing. Additionally, when discussing the results of the OFT it would be informative to include the purpose of AMPH and PCP in the experiment. (i.e. AMPH and/or PCP were administered to increase dopaminergic neurotransmission during the OFT which.....)

Response:

Thank you for your comment. We have rearranged the Results section to improve the flow, particularly in the AMPH and PCP parts. We hope this makes the section easier to read. Additionally, we have included new experiments with cocaine to further complement the results.

7. For the spatial transcriptomics analysis, it would be better to exclude all non-claustrum spots and perform analysis solely on the claustrum. Other spots will exert control over the global and local distancing within the UMAP embedding, and this distorts the relationship between claustrum neurons, the focal point of this paper.

Response:

Thank you for your insightful comment. We appreciate your perspective. We believe that excluding non-claustrum spots would be relevant only if our focus was on identifying CLA subclusters specifically. By conducting clustering analysis without excluding any spots, we

were able to identify the CLA as a transcriptionally distinct cluster, which was not present after the deletion of *Nurr1*. Isolating only claustrum spots could introduce bias into our results.

Furthermore, including all spots in our analysis allows us to demonstrate that the separate cluster observed in AAV-*Nurr1*cKO is unique and does not share transcriptional features with other defined clusters in our sections. This comprehensive approach provides a more accurate and unbiased representation of the data. We hope this clarifies our rationale and the benefits of our methodology.

8. Figure 2h seems to use two different scale bars.

Response:

Great catch. We observed that the same issue was present on Fig. 3 as well. Now we have only one scale bar.

9. There are many other marker genes for claustrum neurons, and neuron subtypes (Erwin et al., eLife 2021). Comparisons to these cell types and transcriptomes, rather than just the genes obtained from the Allen ISH website, would likely be informative.

Response:

Thank you for your excellent recommendation. We are aware of the markers identified in the study by Erwin et al., eLife 2021, and we have noted that the CLA Allen ISH AGEA shares many of the genes presented in that study. In response to your suggestion, we have revised the text in the Figures to show the CLA-specific genes, changing the label from 'AGEA CLA' to 'CLA marker'. Additionally, we have cited the Erwin et al., eLife 2021 study in the Results section, indicating that we took this study into consideration when generating our list of CLA markers. We appreciate your valuable input and hope these modifications improve the clarity and accuracy of our presentation.

Reviewer #3 (Remarks to the Author):

The manuscript by Mantas et al. reports investigations into the developmental regulation of the claustrum (CLA) and the function of the CLA in several behaviors in the mouse. A wide variety of techniques ranging from ISH to electrophysiology to single cell transcriptomics to behavioral pharmacology are performed in the analysis. The manuscript is extremely well written, describing well planned experiments that have resulted in high quality data supporting their

general conclusions. Their overall conclusions support the notion that Nurr1 is critical for the development and function of the claustrum in mice, and that the claustrum is involved in mediating functional connectivity of brain regions similar to that of humans in response to hallucinogens of the psychedelic and KOR classes. Their use of fUS to investigate mouse brain functional connectivity is a strength, demonstrating proof of principle for future use of this technique to investigate connectivity in response to genetic/pharmacological manipulations in rodent brain. Overall the findings and conclusions of the work are of high impact towards our understanding of fundamental brain mechanisms underlying behaviors, and mechanism of action of hallucinogens. I only have a few minor concerns:

DOI should be defined as: 2,5-dimethoxy-4-iodoamphetamine.

Response:

Thank you for clarifying that. We have now replaced the drug's term.

Although the ISH data are convincing that CLA enriched expression is reduced, no antibody confirmation experiments were performed (see Fig 5).

As I am sure the authors are aware, gene expression does not always correlate to protein expression and a disclaimer should be included in the results section of this limitation.

Response:

Great catch. We performed immunofluorescence experiments using a 5HT_{2A}R antibody, which, according to these studies (<https://doi.org/10.1038/npp.2010.195>, <https://doi.org/10.1016/j.ebiom.2016.08.049>), shows a similar immunolabelling pattern to our ISH autoradiographs. We have added a supplementary figure (Supplementary Fig. 5) that demonstrates a robust reduction of 5HT_{2A}R immunolabelling in AAV-Nurr1cKO mice. Given the effectiveness of the antibody, we were eager to investigate the previously described (<https://doi.org/10.1016/j.ebiom.2016.08.049>) DOI-induced 5HT_{2A}R internalization in the CLA of D1R-Nurr1cKO mice. Our observations indicate that D1R-Nurr1cKO mice display a significant reduction of 5HT_{2A}R immunolabelling in the CLA at baseline. DOI (8 mg/kg) treatment resulted in substantial internalization of 5HT_{2A}R in CLA neurons. However, this internalization was markedly reduced in the CLA of D1R-Nurr1cKO mice treated with DOI. These results are now shown in Fig 6.

Is there any speculation as to the mechanism of how Nurr1 activity is involved in specification of the CLA?

Response:

Thank you for this question. While we have some speculations on this matter, further experimentation is required to confirm them. We have included few sentences on this topic in the discussion part.

HTR is largely a reflex of activation of cortical 5-HT_{2A} receptors, and this was unchanged by CLA ablation. A more interesting behavioral experiment to perform would be 2-lever drug discrimination as this probes the introspective cue of psychoactive drugs. Would ablating CLA function in adults leave HTR intact, but remove the ability to discriminate the subjective effects of hallucinogens? Being able to separate these effects in an experimental system would be a major development in the study of psychedelics.*

Response:

Thank you for the excellent recommendation. We are somewhat cautious with the 2-lever drug discrimination test, as the results on κ OR agonists are controversial and may not accurately reflect hallucinatory effects (<https://doi.org/10.1021/cn300138m>, <https://doi.org/10.1007/s00213-008-1458-3>). Additionally, this test is quite sensitive to dopaminergic drugs that promote addiction, such as cocaine, amphetamine, and morphine (<https://doi.org/10.1007/s00213-010-1789-8>, <https://doi.org/10.1007/BF02244073>). κ OR agonists are known to inhibit dopamine release (<https://doi.org/10.1016/j.biopsych.2019.05.012>), which could lead to conflicting results. Since our main focus was to assess effects common among different classes of hallucinogens, we chose to concentrate on fUS recordings. Nonetheless, we have included additional evidence on connectivity effects observed after the ablation of Nurr1 in CLA, using neuropixel recordings on Ctrl, D1R-Nurr1cKO, and AAV-Nurr1cKO mice combined with DOI treatment.

Although not a concern, this is more of a comment. It would be fascinating to use intersectional genetics to express DREADD receptors, especially the KORD, in the CLA for conditional manipulation of activity in adults in behavioral experiments.

Response:

Thank you for your insightful comment. While the idea of using intersectional genetics to express DREADD receptors, particularly KORD, in the CLA for conditional manipulation of

activity in adults is indeed fascinating, we have decided not to pursue this approach in our current study, which is very focused on the role of Nurr1 in the CLA. We appreciate your suggestion and will consider it for future research directions.

Reviewer #4 (Remarks to the Author):

- In this study, Mantas et al., investigated the role of Nurr1 as a central regulator of hallucigenic effect. The study is well conducted, the manuscript well written. The part of the study using ultrasound imaging as a readout for neural correlate is well thoughts and statistical analysis seem appropriate. There are, however, several points (listed below) to consider in order to strengthen and clarify the study.

- Authors should state more clearly what is the rational for the use of ultrasound imaging in their study. This technique of ultrafast Doppler can be used to measure different modalities. They introduce the concept of “neuronal correlate”. It is true that due to neurovascular coupling, when animals are exposed to a task, changes in CBV are due to increased neuronal activation. However, this is not always the case during a task, as recently review by Patrick Drew (<https://doi.org/10.1016/j.tins.2022.08.004>). In addition, in pharmacological studies, things are far more complicated. Pharmacologically active drugs may have a direct unspecific vascular effect, which has nothing to do with neural correlate; or can be in addition to it.

In addition, here the study does not use functional ultrasound, as they did not do any activations per se. They rather performed pharmacological fUS, which is different. This only appears in the discussion section.

Authors should mention these concepts, correct and clarify their text accordingly.

Response:

Thank you for your valuable contribution. In response to your suggestions, we have replaced the term 'neuronal correlate' and updated 'fUS' to 'pfUS' (pharmacological functional ultrasound) where appropriate. Additionally, we have included a discussion on the limitations of pfUS due to the direct effects of the drugs on blood vessels. We have also added experiments with neuropixel recordings in Figure 7, demonstrating that pfUS does not necessarily imply neuronal activity, as you mentioned. We hope these modifications address your concerns comprehensively.

In addition, authors should quantify the unspecific vascular effect induced by the drugs tested, using variations of CBV in the medial vein and in one artery. They are both clearly visible on panel C of figure 7.

Response:

Thank you for your thoughtful suggestion. We truly appreciate the idea of quantifying the nonspecific vascular effects induced by the drugs by measuring variations in cerebral blood volume (CBV) in the medial vein and a specific artery. However, we encountered significant variability in the visibility of these structures across different brain samples, which could introduce bias when defining the regions of interest (ROIs). To address this concern, we utilized the Allen Brain Reference Atlas Registration tool to select brain ROIs for our analysis. This approach effectively eliminates potential bias in ROI selection, ensuring that our results are consistent and reliable. Unfortunately, since we cannot use the registration tool for the vessels, we are unable to perform a similar analysis with the same level of consistency. We hope this explanation clarifies our decision and the challenges involved. Thank you once again for your valuable input.

- One major issue of this study is the low Doppler signal in the anterior part of the brain in the adult mice, especially in the lateral part. It is very difficult and scientifically inaccurate to study changes if the signal is weak/inexistent. This is particularly problematic for the SM that seems to have no signal at all. There is no problem in medial structures, however.

Response:

Thank you for your keen observation. We believe that the signal is not absent, but rather that there are no major vessels running through the brain at these coronal levels. This is evident in our supplementary Figure 11, particularly in Supplementary Figure 11c, where we assess the pfUS sedation protocol. You can observe signal activity in the very lateral parts of the brain at this coronal level. In this experiment, we measured the signal from the somatosensory cortex (SM) during lower limb stimulation, which showed a strong increase in CBV in response to the sensory stimulus. We hope this explanation adequately addresses your concerns.

- What is the rationale for the choice of regions of interest to study? In particular, why use overlapping areas? Only at the end of the discussion, the reader understands your motivation: to compare in your animal model what other teams described in human (decreased CBV in

some of these ROI and increased functional connectivity between PFC and SM). This should be stated earlier in the manuscript.

Response:

Thank you for the excellent recommendation! We have added relevant text at the end of the introduction to introduce the concept of our recordings. Additionally, we have clarified the rationale behind our approach at the beginning of the ‘Nurr1 deletion in the CLA alters cortical functional connectivity in response to hallucinogen receptor agonists’ section of the Results.

Minor points

- Some of your figures or figure legends are not accurate enough. For instance:
- It is unclear from the figure legend if the results presented in panel E are an example or an average over X animals.

Response:

Thank you for your observation. We have added text to clarify that the numbers in the heatmaps represent the average of the animals. Additionally, we have revised the figure legend to enhance its clarity.

- What is the X axis of the panel E? Is it time? This is not clear.

Response:

Great catch! Yes, the x-axis indeed represents time. We have corrected the figure to make this clearer. Thank you for pointing it out.

- In panels f and g: Why is there a pink rectangle with the mention ‘PFC’? The reviewer understood that these were the changes of CBV in the ACA(f) and MO (g). But not in the PFC. This is unclear.

Response:

Thank you for your valuable recommendations. We have modified Figure 7 to enhance its readability and clarity for the reader. We hope these changes address your concerns effectively.

REVIEWER COMMENTS

Reviewer #1 (Remarks to the Author):

The authors should be commended for an attentive revision. The paper is now much stronger. Two issues remain, one major, one minor:

Major: Thank you for including the behavior data as part of Figure 4i-n. Despite previous claims that claustrum is involved in these tasks, the fact remains that there is still no positive control: claustrum disruption is not shown to alter behavior in these tasks in your hands.

Minor: I appreciate the move to call the claustrum/dorsal endopiriform the claustrum complex, especially given previous publications doing the same. For accuracy sake, and to not suggest to readers that the endopiriform is part of claustrum (they clearly are different nuclei based on connectivity), I would suggest claustrum/dorsal endopiriform nucleus complex.

Reviewer #2 (Remarks to the Author):

The authors have done an excellent job responding to my comments and revising their manuscript. I commend the authors for their great paper here and support publication in its current form.

Reviewer #3 (Remarks to the Author):

Each of my comments has been sufficiently addressed in the revision.

There are 2 minor typos:

- 1) line 187 - the word "destructor" is used. Perhaps they meant distractor?
- 2) References 37 and 38 are duplicates of one another.

Reviewer #4 (Remarks to the Author):

The authors have corrected the manuscript satisfactorily.

The new wording 'pfUS' (instead of fUS) is entirely appropriate.

In addition, in order to strengthen the pfUS study, they added new neuropixel experiments which provide very interesting concordant results.

Response to Referees

Reviewer #1 (Remarks to the Author):

The authors should be commended for an attentive revision. The paper is now much stronger. Two issues remain, one major, one minor:

Major: Thank you for including the behaviour data as part of Figure 4i-n. Despite previous claims that claustrum is involved in these tasks, the fact remains that there is still no positive control: claustrum disruption is not shown to alter behavior in these tasks in your hands.

Response:

Thank you for your insightful comment. We acknowledge the absence of a positive control (claustrum-specific inhibition) in our behavioural studies, which would indeed strengthen our conclusions. However, our study's primary focus is on the effects of Nurr1 deletion rather than the general function of the CLA. Our aim was to characterize the mouse line we generated and present the behavioural data accordingly. We recognize that the role of CLA in behaviour is a developing field with no clear consensus within the scientific community. Existing studies often present contrasting effects in the same behavioural assays, such as those examining anxiety or depressive-like behaviours (DOI:10.1126/sciadv.abi6375 and <https://doi.org/10.1038/s41467-023-43636-x>). Another example that highlights the complexity and variability of behavioural readouts associated with CLA inhibition is the the studies that emphasize on CLA's role in slow wave sleep. While it is reproducible that CLA's role in slow-wave sleep electrophysiology is important (<https://doi.org/10.1038/s41586-020-1993-6>, <https://doi.org/10.1038/s41593-020-0625-7>, <https://doi.org/10.1038/s41467-024-48829-6>, <https://doi.org/10.1016/j.celrep.2023.113620>) CLA-specific ablation does not appear to affect sleep duration (<https://doi.org/10.1038/s41593-020-0625-7>). We also considered the methodological limitations of intersectional approaches, such as using retroAAV Cre in cortical areas combined with a Cre-dependent neuronal silencing vector in the CLA, which target specific projection neurons rather than the entire CLA structure. Another viable approach involves using CLA-specific transgenic mouse lines (Gnb4-Cre), which we currently do not have access to in our lab (<https://doi.org/10.1016/j.neuron.2022.10.026>). We have added a paragraph in the discussion section addressing these points and the broader context of our findings. We hope this addition clarifies our rationale and addresses your concerns. Thank you again for your valuable feedback.

Minor: I appreciate the move to call the claustrum/dorsal endopiriform the claustrum complex, especially given previous publications doing the same. For accuracy sake, and to not suggest to readers that the endopiriform is part of claustrum (they clearly are different nuclei based on connectivity), I would suggest claustrum/dorsal endopiriform nucleus complex.

Response:

Thank you for your suggestion. We have updated the terminology to "claustrum/dorsal endopiriform nucleus complex" in the title and the text to ensure accuracy and clarity regarding the distinctiveness of these nuclei.

Reviewer #2 (Remarks to the Author):

The authors have done an excellent job responding to my comments and revising their manuscript. I commend the authors for their great paper here and support publication in its current form.

Response:

Thank you very much for your kind words and support. We are delighted that you find the revisions satisfactory and appreciate your positive feedback on our paper.

Reviewer #3 (Remarks to the Author):

Each of my comments has been sufficiently addressed in the revision.

There are 2 minor typos:

- 1) line 187 - the word "destructor" is used. Perhaps they meant distractor?
- 2) References 37 and 38 are duplicates of one another.

Response:

Thank you for your feedback and for pointing out these typos. We have corrected "destructor" to "distractor" on line 187. Additionally, we have removed the duplicate reference, ensuring that references 37 and 38 are unique.

Reviewer #4 (Remarks to the Author):

The authors have corrected the manuscript satisfactorily.

The new wording 'pfUS' (instead of fUS) is entirely appropriate.

In addition, in order to strengthen the pfUS study, they added new neuropixel experiments which provide very interesting concordant results.

Response:

Thank you for your constructive comments. We are pleased that the changes, including the new wording 'pfUS' and the addition of the neuropixel experiments, have met your approval and strengthened the study.

REVIEWERS' COMMENTS

Reviewer #1 (Remarks to the Author):

I am satisfied by the authors' response. Publish.